# Compensatory induction of MYC expression by sustained CDK9 inhibition via a BRD4-dependent mechanism

Huasong Lu[1,2†], Yuhua Xue[2†], Guoying K Yu[3], Carolina Arias[3], Julie Lin[3], Susan Fong[3], Michel Faure[3], Ben Weisburd[3], Xiaodan Ji[1], Alexandre Mercier[3], James Sutton[3], Kunxin Luo[1], Zhenhai Gao[3]*, Qiang Zhou[1]*

[1]Department of Molecular and Cell Biology, University of California, Berkeley, Berkeley, United States; [2]Innovation Center of Cell Signaling Network, School of Pharmaceutical Sciences, Xiamen University, Xiamen, China; [3]Novartis Institute for BioMedical Research, Emeryville, United States

*For correspondence: zgao@incyte.com (ZG); qzhou@berkeley.edu (QZ)

†These authors contributed equally to this work

Competing interests: The authors declare that no competing interests exist.

**Abstract** CDK9 is the kinase subunit of positive transcription elongation factor b (P-TEFb) that enables RNA polymerase (Pol) II's transition from promoter-proximal pausing to productive elongation. Although considerable interest exists in CDK9 as a therapeutic target, little progress has been made due to lack of highly selective inhibitors. Here, we describe the development of i-CDK9 as such an inhibitor that potently suppresses CDK9 phosphorylation of substrates and causes genome-wide Pol II pausing. While most genes experience reduced expression, MYC and other primary response genes increase expression upon sustained i-CDK9 treatment. Essential for this increase, the bromodomain protein BRD4 captures P-TEFb from 7SK snRNP to deliver to target genes and also enhances CDK9's activity and resistance to inhibition. Because the i-CDK9-induced MYC expression and binding to P-TEFb compensate for P-TEFb's loss of activity, only simultaneously inhibiting CDK9 and MYC/BRD4 can efficiently induce growth arrest and apoptosis of cancer cells, suggesting the potential of a combinatorial treatment strategy.

## Introduction

The proper control of eukaryotic gene expression is fundamental for normal development and cellular response to environmental challenges. The control frequently occurs at the level of transcription, where RNA polymerase (Pol) II is employed to execute a series of interconnected stages that collectively constitute the transcription cycle. In the past, the early stages of this cycle involving the recruitment of Pol II to gene promoters and assembly of active pre-initiation complexes were considered the primary step where transcription is controlled (*Kuras and Struhl, 1999*; *Ptashne, 2005*). However, recent evidence indicates that the subsequent stages are also frequently targeted to regulate gene expression. For example, genome-wide analyses from *Drosophila* to mammals have shown that promoter-proximal pausing of Pol II is a prevalent feature of many genes and that the regulated release of Pol II is essential for synchrony and robustness of their induction (*Guenther et al., 2007*; *Muse et al., 2007*; *Zeitlinger et al., 2007*; *Levine, 2011*; *Zhou et al., 2012*).

During transcription, the extensive and dynamic modifications of the Pol II C-terminal domain (CTD) have been linked to specific stages of the transcription cycle and mRNA processing. Among these, the CTD Serine-2 phosphorylation, which is a hallmark of productive elongation and RNA processing, is catalyzed by the positive transcription elongation factor b (P-TEFb), which is composed of CDK9 and its cyclin partner T1 (CycT1) or the minor forms T2a and T2b. Additionally, P-TEFb also phosphorylates the SPT5 subunit of DSIF and the NelfE subunit of NELF, which antagonizes the

**eLife digest** Cancers are often caused by mutations in genes that allow cells to proliferate uncontrollably. One gene that is frequently mutated in many cancers encodes a protein called MYC. The activity of this gene is normally tightly controlled, but the mutations found in human cancer cells mean that this gene is constantly switched on, and so too much MYC protein is produced.

Previous studies have shown that a protein complex called 'positive transcription elongation factor b' (or P-TEFb for short) is essential to control the expression of the gene for MYC. P-TEFb works with an enzyme called RNA polymerase II to copy the instructions contained in protein-coding genes into long molecules called messenger RNAs. This process is called transcription and it involves a number of stages. P-TEFb is needed to start of one these stages, which is known as the 'elongation' step.

P-TEFb consists of two protein subunits; one of which—an enzyme called CDK9—is the catalytic subunit. Most of the P-TEFb complexes in a cell are held in an inactive form, in which the activity of the CDK9 subunit is suppressed. If CDK9 is active when it should not be, certain proteins—including the MYC protein—can be produced in abnormally high amounts. This means that inhibiting CDK9 has been investigated as one way to reduce the production of the MYC protein. While some CDK9 inhibitors already exist, these compounds have the undesirable effect of also inhibiting other related enzymes and thus killing normal cells. Hence, new and more selective inhibitors of CDK9 are urgently needed.

Lu, Xue et al. have now developed a new inhibitor of CDK9, called i-CDK9. The experiments show that i-CDK9 can potently inhibit CDK9 activity; and in human cells, very low levels of i-CDK9 prevented RNA polymerase II carrying out elongation for many genes. Unexpectedly, Lu, Xue et al. observed that more messenger RNA molecules that encode MYC were produced after cells were treated with low levels of i-CDK9. Further investigation revealed that this unexpected result occurred because the P-TEFb complexes were released from the inactive form and brought to the MYC gene by another protein called BRD4. This stimulated production of the MYC messenger RNAs. When P-TEFb was bound by BRD4, the CDK9 activity was also protected against inhibition by i-CDK9. Moreover, the reason that the MYC expression was induced by i-CDK9 is because the cells can compensate for the loss of CDK9 by using MYC to maintain the production of messenger RNAs of many key genes; these genes include the gene for MYC itself. These results suggest that CDK9 and MYC must be simultaneously inhibited in order to effectively treat cancers.

inhibitory actions of these two negative elongation factors and promotes the release of paused Pol II and transition into productive elongation (*Zhou et al., 2012*).

The importance of P-TEFb in transcriptional elongation requires that its activity be tightly controlled in the cell. Indeed, under normal growth conditions, the majority of P-TEFb is sequestered in the inactive 7SK snRNP, in which the CDK9 kinase activity is suppressed by HEXIM1 or 2 in a 7SK snRNA-dependent manner (*Nguyen et al., 2001*; *Yang et al., 2001*; *Yik et al., 2003*). The remaining P-TEFb is catalytically active and present in a BRD4-containing complex and the super elongation complex (SEC) (*Zhou et al., 2012*). In the former, the BET bromodomain protein BRD4 serves to recruit P-TEFb to the promoters of many primary response genes (PRGs) through binding to acetylated chromatin or the transcriptional mediator complex (*Jang et al., 2005*; *Yang et al., 2005*, *2008*). The SEC, on the other hand, is a target of the Tat protein encoded by the HIV-1 virus or the MLL (mixed lineage leukemia) fusion proteins created by chromosomal translocations to stimulate transcriptional elongation of HIV-1 and MLL-target genes, respectively (*Mueller et al., 2009*; *He et al., 2010*; *Lin et al., 2010*; *Sobhian et al., 2010*; *Yokoyama et al., 2010*).

A number of reagents and conditions that can globally impact growth and/or induce stress response have been shown to cause the release of P-TEFb from 7SK snRNP and formation of the BRD4-P-TEFb complex for stimulation of transcriptional elongation (*Zhou and Yik, 2006*; *Zhou et al., 2012*). In HIV-1 infected cells, however, Tat has been shown to directly extract P-TEFb from 7SK snRNP to assemble the Tat-SEC complex on the viral promoter (*Barboric et al., 2007*; *Sedore et al., 2007*; *Lu et al., 2014*). Multiple lines of evidence support the notion that the 7SK snRNP represents a cellular reservoir of unused P-TEFb activity, which can be withdrawn in response to

various signals to form active P-TEFb complexes for activation of cellular and viral genes (*Zhou et al., 2012*; *Lu et al., 2013*).

The proto-oncogene *MYC* occupies a central position downstream of many growth-promoting signal transduction pathways. As an immediate early response gene activated by many membrane-associated ligand–receptor complexes, MYC links growth factor stimulation to regulated cellular proliferation and cell cycle progression under normal conditions (*Levens, 2013*). Because of this property, it is one of the most frequently amplified genes in tumors, a major genetic change that leads to uncontrolled proliferation of cancer cells (*Dang, 2012*).

The expression of MYC is normally controlled at almost every possible level to achieve a proper concentration of the protein for optimal cell growth. Prior to our current understanding of the pervasiveness of elongation control, MYC was in fact one of the first few cellular genes found to be regulated by this particular mechanism (*Bentley and Groudine, 1986*). Recent studies employing the BET bromodomain inhibitors such as JQ1 and iBET-151 have provided fresh mechanistic insights into how this is accomplished (*Filippakopoulos and Knapp, 2014*; *Shi and Vakoc, 2014*). The published data indicate that BRD4 and its recruitment of P-TEFb to the MYC promoter region play a particularly important role in MYC expression in human cancer cells. JQ1 and iBET-151 are shown to bind to the BRD4 bromodomains and displace the BRD4-P-TEFb complex from acetylated chromatin to inhibit MYC transcription, which in turn induces differentiation and growth arrest of cancer cells (e.g., acute myeloid leukemia, multiple myeloma, and Burkitt's lymphoma) that are addicted to MYC (*Filippakopoulos and Knapp, 2014*; *Shi and Vakoc, 2014*).

Because the inhibition of BRD4-mediated recruitment P-TEFb to MYC and other PRGs by JQ1 and iBET-151 can suppress cancer growth and progression, it is tempting to speculate that the direct inhibition of P-TEFb itself will produce a similar or perhaps even more focused effect. In support of this idea, a recent study implicates CDK9 inhibition as an effective therapeutic strategy for MYC-overexpressing liver tumors (*Huang et al., 2014*). Although small molecule pan CDK inhibitors (e.g., flavopiridol and SNS-032) with potent anti-CDK9 activities already exist and have been valuable tools for exploring the functions of P-TEFb (*Chao et al., 2000*; *Chen et al., 2009*), they display pleiotropic effects indicative of interference with other cellular enzymes/pathways (*Bible et al., 2000*). At this moment, there are few verified and well-characterized compounds that show high selectivity and potent inhibitory activity against CDK9.

In light of the tremendous interest in P-TEFb as a potential therapeutic target, we describe here the development and characterization of a potent and selective CDK9 inhibitor called i-CDK9 that efficiently suppresses P-TEFb's phosphorylation of the Pol II CTD and the DSIF subunit SPT5 and causes widespread Pol II pausing at gene promoters. Although the vast majority of genes display a drastic i-CDK9-induced reduction in gene expression, a small group of genes show a surprising increase in expression after the drug treatment, and the proto-oncogene MYC and several other key PRGs are among them. In the current study, we explore the molecular mechanism as well as biological significance of i-CDK9-induced MYC expression. Our data reveal the essential dual roles played by BRD4 in MYC induction and indicate that the elevated expression in i-CDK9-treated cells is part of the cellular compensation for the loss of CDK9. Because of this compensatory mechanism, our data demonstrate that the simultaneous inhibition of both CDK9's catalytic activity and MYC's expression or function causes synergistic induction of growth arrest and apoptosis of cancer cells.

## Results

### i-CDK9 is a potent and selective CDK9 inhibitor

High throughput screening combined with crystal structure-enabled lead compound optimization has led to the identification of a novel and potent CDK9-selective inhibitor called i-CDK9 (the screen and a co-crystal structure of i-CDK9 bound to CDK9 will be described elsewhere). i-CDK9 has a N2′-(trans-4-aminocyclohexyl)-5′-chloro-N6-(3-fluorobenzyl)-2,4′-bipyridine-2′,6-diamine scaffold that is structurally distinct from flavopiridol and all the other known non-selective CDK inhibitors (*Figure 1A*). It occupies the ATP-binding pocket of the CDK9 kinase domain as revealed by a co-crystal structure solved at 2.6 Å resolution (manuscript in preparation).

Using an in vitro AlphaScreen (PerkinElmer, Inc)-based kinase assay, i-CDK9 was shown to potently inhibit the CDK9-CycT1 catalytic activity with an $IC_{50}$ value below the detection limit of 0.0004 μM (*Figure 1B*). In the same assay, flavopiridol showed an $IC_{50}$ value of 0.007 μM. Furthermore, compared

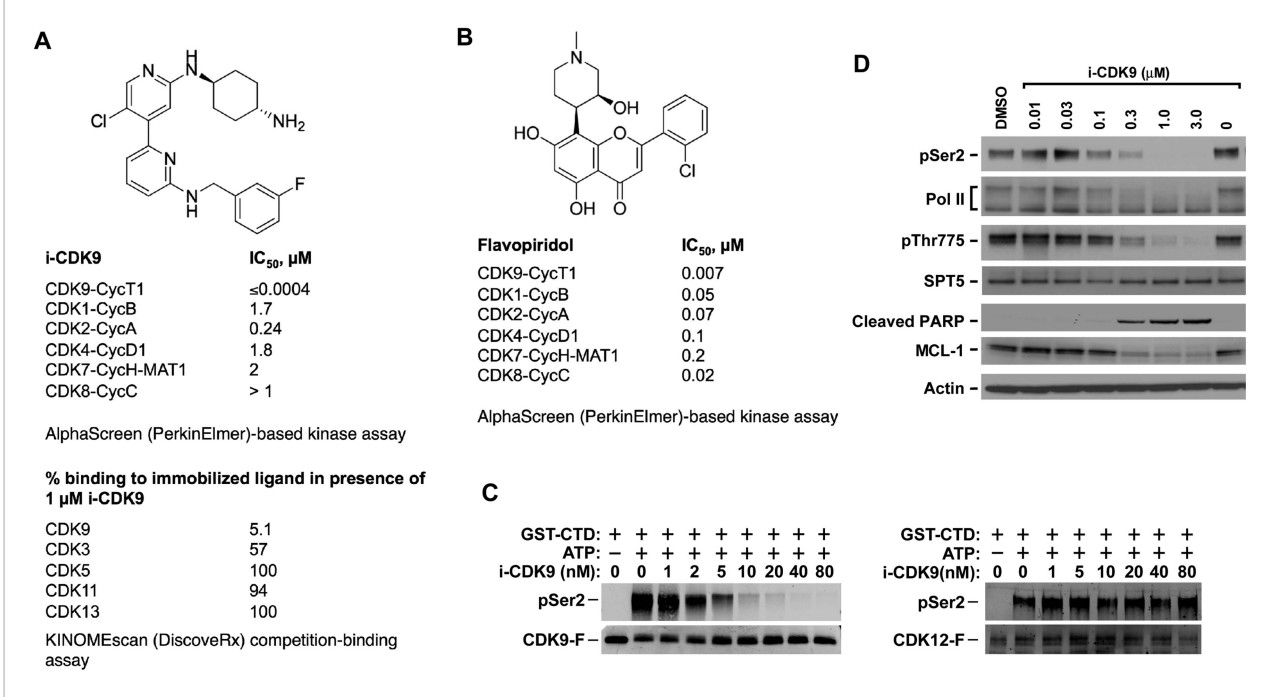

**Figure 1**. i-CDK9 is a potent and selective CDK9 inhibitor that elicits cellular responses indicative of P-TEFb inhibition. (**A** and **B**) Structures and selectivity profiles of i-CDK9 (**A**) and falvopiridol (**B**). The numbers refer to the concentrations (μM) of the two compounds that resulted in 50% inhibition of the enzymatic activity of the indicated CDK–cyclin pairs in the AlphaScreen (PerkinElmer)-based kinase assay (**A**) or 50% inhibition of the bindings of the indicated CDKs to the immobilized ligands in the KINOMEscan platform (**B**). (**C**) In vitro kinase reactions containing affinity-purified CDK9-F-CycT or CDK12-F-CycK and GST-CTD as a substrate were conducted in the presence of the indicated concentrations of i-CDK9. pSer2 and the Flag-tagged kinase in each reaction were detected by Western blotting with anti-pSer2 and anti-Flag antibodies, respectively. (**D**) HeLa cells were treated for 8 hr with DMSO or the indicated concentrations of i-CDK9. Total cell lysates were examined by immunoblotting for the proteins labeled on the left.

The following source data and figure supplement are available for figure 1:

**Source data 1**. Selectivity profile of i-CDK9.

**Figure supplement 1**. Recombinant CDK12-CycK is less sensitive to inhibition by i-CDK9.

to its inhibition of the CDK9-CycT1 kinase activity, i-CDK9 exhibited at least 600-fold lower activity toward CDK1-CycB, CDK2-CycA, CDK4-CycD1, CDK7-CycH-MAT1 and CDK8-CycC (*Figure 1B*). In contrast, flavopiridol displayed only 2.9- to 28.6-fold lower activity toward the same CDK-cyclin pairs compared to CDK9-CycT1 (*Figure 1B*).

As there are no commercial or in-house kinase assays available for CDK3, CDK5, CDK11 and CDK13, the inhibitory abilities of i-CDK9 toward these CDKs were thus evaluated in the DiscoveRx KINOMEscan assay, which is based on a proprietary active site-directed competition-binding platform (*Fabian et al., 2005*). At 1 μM, i-CDK9 almost completely blocked the binding of CDK9 to the immobilized ligands (only 5.1% CDK9 captured on solid support; *Figure 1A*), but displayed essentially no effect on CDK5 (100% binding to immobilized ligands), CDK11 (94%), CDK13 (100%) and only a partial inhibition of the binding of CDK3 (57%).

Among all members of the CDK super family, CDK12 deserves special attention because of its reported phosphorylation of the Pol II CTD on Ser2, an ability that is shared with CDK9 (*Bartkowiak et al., 2010*). To determine whether i-CDK9 also affects CDK12 kinase activity, we examined the abilities of affinity-purified Flag-tagged CDK12 (CDK12-F), CDK9 (CDK9-F) and their associated cyclin partners to phosphorylate GST-CTD in the presence of increasing amounts of i-CDK9. Phosphorylation of Ser2 (pSer2) was detected by Western blotting with a specific antibody. While Ser2 phosphorylation by CDK9-F was efficiently inhibited by i-CDK9 with an estimated IC$_{50}$ of ~2 nM under the current

experimental conditions, no obvious inhibition of CDK12-F was detected even at 80 nM of the inhibitor (*Figure 1C*).

To confirm this result using materials from a different source, the sensitivity of the two kinases to i-CDK9 was also compared in reactions containing the baculovirus-produced recombinant CDK9-CycT1 and CDK12-CycK (SignalChem). While Ser2 phosphorylation by CDK9-CycT1 was mostly inhibited by 80 nM i-CDK9, CDK12-CycK was not significantly inhibited until 640-1200 nM i-CDK9 was added into the reactions (*Figure 1—figure supplement 1*). Thus, between CDK9 and CDK12, i-CDK9 displayed markedly higher selectivity against the former. It is interesting to note that a similar finding has also been made with the pan-CDK inhibitor flavopiridol (*Bosken et al., 2014*).

Finally, the selectivity profile of i-CDK9 toward other non-CDK kinases was also assessed using the KINOMEscan platform. Among the >400 kinases evaluated, only 9 (highlighted yellow in *Figure 1—source data 1*) consistently showed less than 40% binding to the immobilized ligands in the presence of 1 or 10 μM of i-CDK9. Except for CLK4, for which no suitable assay was available, the rest of the kinases plus a selected group of others were re-examined for the concentration-dependent effect of i-CDK9 in either functional kinase assays or binding assays. Again, i-CDK9 was found to display great selectivity for CDK9 with more than 100-fold difference detected between CDK9 and the next two best targets DYRK1A and DYRK1B (IC50 < 0.0004 μM for CDK9 vs IC50 = 0.055 and 0.047 for DYRK1A and B, respectively; column D in *Figure 1—source data 1*). The fact that DYRK1A can also be inhibited by i-CDK9 albeit with reduced efficiency suggests that the structure and/or function of this kinase may resemble that of CDK9 to a certain degree. Indeed, it has recently been reported that DYRK1A acts as a Pol II CTD Kinase at its target gene promoters (*Di Vona et al., 2015*). In contrast to i-CDK9, flavopiridol displayed strong but non-selective interactions with a far greater number of kinases in the DiscoveRx panel (see http://www.discoverx.com/ReferenceTreeImages/Flavopiridol.htm), some of which (e.g., with ICK, CDK4-CycD1 and CDKL5) had affinities that are similar or even higher than that for CDK9.

## i-CDK9 elicits cellular responses indicative of P-TEFb inhibition

Consistent with its specific binding to and potent inhibition of purified CDK9-CycT1 in vitro, i-CDK9 markedly reduced the CDK9-mediated pSer2 in the Pol II CTD and Thr775 (pThr775) in the DSIF subunit SPT5 in a dose-dependent manner in HeLa cells (*Figure 1D*). Inhibition of phosphorylation of both Pol II CTD and SPT5 is expected to block transcriptional elongation, which will effectively decrease the production of labile proteins from short-lived transcripts. Indeed, correlating with the decrease in cellular levels of pSer2 and pThr775, i-CDK9 markedly down-regulated the expression of the short-lived anti-apoptotic protein MCL-1 and at the same time induced proteolytic cleavage of poly(ADP-ribose) polymerase (PARP), which is considered a major hallmark of apoptosis and caspase activation (*Figure 1D*). Collectively, these data demonstrate that i-CDK9 is a highly potent and selective inhibitor of CDK9 that is able to elicit cellular responses indicative of P-TEFb inhibition.

## Inhibition of CDK9 causes Pol II pausing at numerous gene promoters

To investigate the global impact of CDK9 inhibition on transcriptional elongation by Pol II, chromatin immunoprecipitation followed by massively parallel DNA sequencing (ChIP-seq) was performed to examine the genome-wide occupancy of Pol II before and after HeLa cells were treated with i-CDK9. Similar to the situation reported in other cell types (*Zeitlinger et al., 2007*; *Rahl et al., 2010*; *Liu et al., 2013*), among the total of 11,197 genes with detectable Pol peaks, 7805 (70%) showed a traveling ratio (TR; also called the pausing index; (*Zeitlinger et al., 2007*; *Rahl et al., 2010*) greater than 2.0 in the control DMSO-treated HeLa cells. Since TR is defined as the relative ratio of Pol II density in the promoter-proximal region vs the gene body (*Figure 2A*), the above numbers suggest that the majority of the genes were experiencing promoter-proximal pausing by Pol II in the control cells. Importantly, upon treatment with i-CDK9 for 2 and 8 hr, 6339 (56.6%) and 7658 (68.4%) genes displayed an increase of at least 1.5-fold in Pol II TR, respectively (*Figure 2B*), indicating significantly elevated promoter-proximal pausing by Pol II upon CDK9 inhibition. *Figure 2C* shows four representative genes (SMUG1, TMEM115, SEC13 and CSNK1D) with significantly elevated Pol II TR upon 8 hr of i-CDK9 treatment.

To investigate how the changes in TR value may correlate with changes in expression of the genes, we conducted DNA microarray studies to determine the effect of selective CDK9 inhibition on global

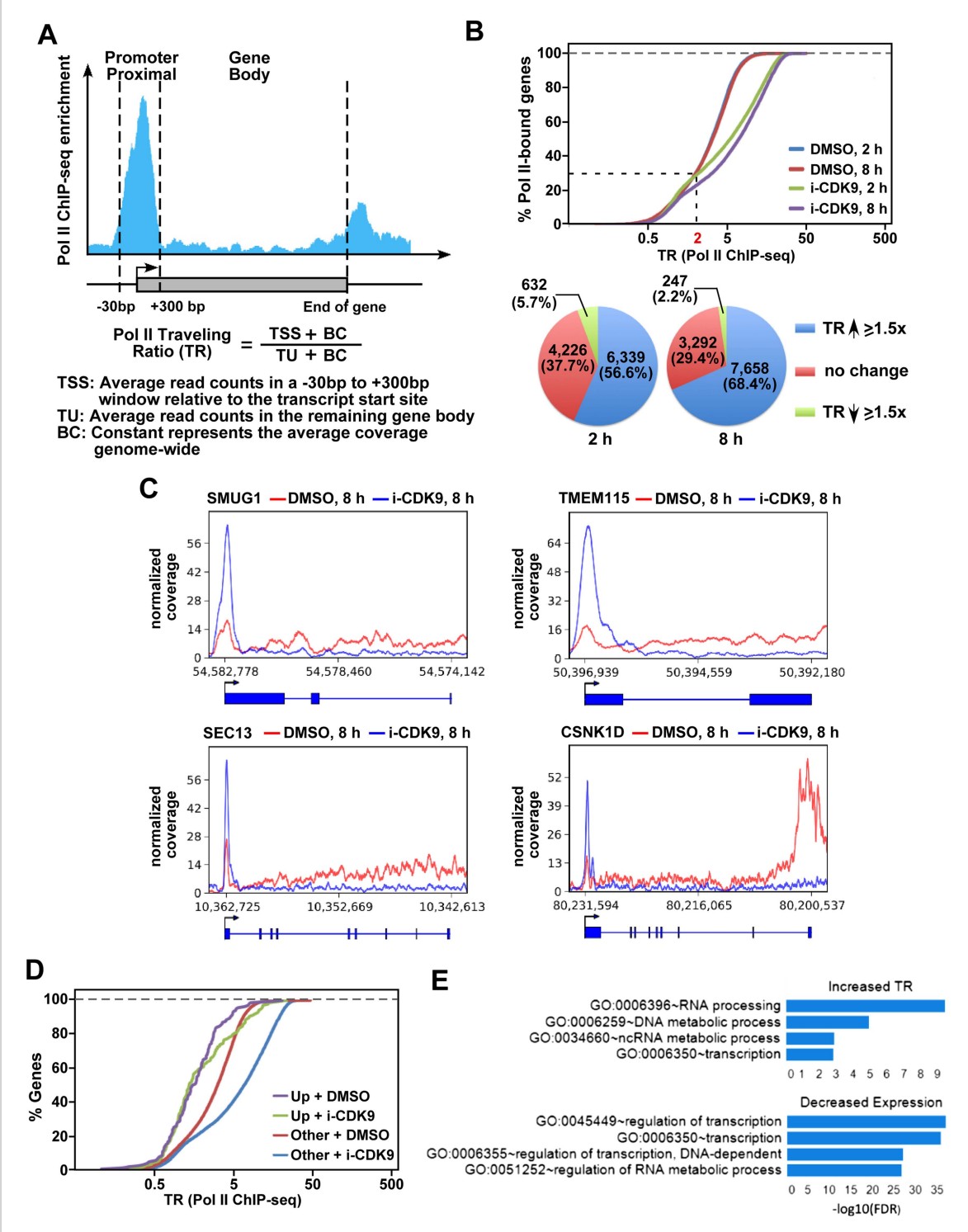

**Figure 2**. i-CDK9 causes widespread promoter-proximal pausing by Pol II and the biggest decrease in expression for genes involved in regulation of transcription and RNA metabolic process. (**A**) Schematic diagram illustrating calculation of the Pol II traveling ratio (TR). (**B**) Distribution of Pol II-bound genes with a given TR as determined by ChIP-seq under the various conditions as indicated. The pie charts below describe the percentages of genes with 1.5-fold increase, 1.5-fold decrease, or no change in TR after exposure to i-CDK9 for 2 or 8 hr as compared to DMSO. (**C**) Occupancy of Pol II as revealed by ChIP-seq across 4 representative genes with increased TR after CDK9 inhibition. The read coverage is shown for the entire gene plus a margin on either side equal to 7% of the gene length. (**D**) Distribution of Pol II-bound genes with a given TR as determined by ChIP-seq. The genes are grouped by expression changes induced by i-CDK9. Up: the 138 genes that showed at least twofold increase in expression after exposure to i-CDK9 or 8 hr. Other:
*Figure 2. continued on next page*

Figure 2. Continued

genes whose expression was either unaffected or affected less than twofold by i-CDK9. (E) Enrichment of GO biological processes by DAVID. Only top 4 gene sets are shown for top 500 genes with the biggest increase in TR at 8 hr treatment with i-CDK9 (top) and top 500 genes with the largest decrease in gene expression at 8 hr i-CDK9 treatment (bottom).

gene transcription. A comparison between the two data sets indicates that TR values for those 138 genes whose expression was up-regulated at least twofold by i-CDK9 were significantly less affected compared to the rest of the genes (*Figure 2D*). Furthermore, the top 4 gene sets with the largest increase in TR show an enrichment in biological processes ranging from RNA processing, DNA and non-coding RNA metabolic process to transcription (*Figure 2E*). Largely consistent with this TR-based analysis, the top 4 sets, whose expression had the biggest drop upon i-CDK9 treatment, are populated by genes involved in regulation of transcription and RNA metabolic process, indicating that these processes and their control are particularly susceptible to inhibition of CDK9.

## Induction of MYC expression in response to sustained inhibition of CDK9

Among the small group of genes that showed at least twofold increase in expression after 8 hr of i-CDK9 treatment, the proto-oncogene MYC has caught our special attention because of its extreme importance in cell growth control and oncogenic transformation and also the fact that its expression has been shown to depend on CDK9 (*Kanazawa et al., 2003*; *Rahl et al., 2010*; *Huang et al., 2014*). Microarray analysis performed at three different time points (2, 8 and 16 hr) indicates that MYC expression in i-CDK9-treated HeLa cells displayed an interesting biphasic responses with a small initial decrease (within 2 hr) followed by a dramatic rebound (around 8 hr and continue at 16 hr) in the presence of continuous CDK9 inhibition by i-CDK9 (*Figure 3A*). This result was further confirmed by qRT-PCR analysis of MYC mRNA levels in cells treated with i-CDK9 or DMSO (*Figure 3—figure supplement 1*).

In addition to HeLa cells, a similar biphasic expression pattern was also observed in four other cell lines representing a wide spectrum of human cancers and *MYC* amplification states (*Figure 3A*; http://www. broadinstitute.org/ccle/home). Notably, the extent of the initial decrease and then subsequent induction of MYC mRNA production varied among the cell lines. For example, after 2 hr of i-CDK9 (0.5 μM) treatment, the MYC mRNA levels in U87MG (*MYC* copy number [CN] = 2) and HeLa (CN = 4) cells showed only a slight decrease compared to the DMSO control, whereas they were significantly reduced in A375 (CN = 2), NCIH441 (CN = 8), and A2058 (CN = 4) cells. Furthermore, the prolonged treatment for 8–16 hr caused the MYC mRNA to reach levels much higher than those in the control cells in all cell lines except NCIH441 (*Figure 3A*). Despite these variations, which could well be caused by differences in MYC mRNA stability in difference cell lines, it is clear that the induction of MYC expression in response to prolonged CDK9 inhibition by i-CDK9 is a general phenomenon likely caused by a common mechanism independent of the MYC amplification/expression levels. In contrast to MYC, another short-lived gene HEXIM1, which was previously shown to depend on active CDK9 for expression (*He et al., 2006*), was continuously suppressed throughout the entire 16 hr of i-CDK9 treatment in all five cell lines (*Figure 3—figure supplement 2*).

## i-CDK9 induces MYC expression before global phosphorylation of Pol II CTD on Ser2 is completely shut down

How can the sustained global inhibition of P-TEFb, a general transcription elongation factor, cause such a surprising and dramatic induction in MYC's expression? We decided to use HeLa cells, which are a convenient and reliable model system for conducting many biochemical analyses, to determine the molecular mechanism behind this phenomenon. While most of the experiments were done in this cell line, a few key conclusions were also confirmed in other cell types (see below). As the initial reduction in MYC mRNA level was not prominent in HeLa cells, we focused our attention on the major up-regulation of MYC expression detected upon 8–16 hr of treatment with i-CDK9.

First, the induction of MYC protein level was confirmed in HeLa cells treated with i-CDK9. Upon exposure to just 0.1 μM i-CDK9, the MYC level increased 3.6-fold whereas the global pSer2 level was only decreased by 23% (*Figure 3B*). It took 1.0 μM i-CDK9 to cause up to 85% reduction in pSer2. In a

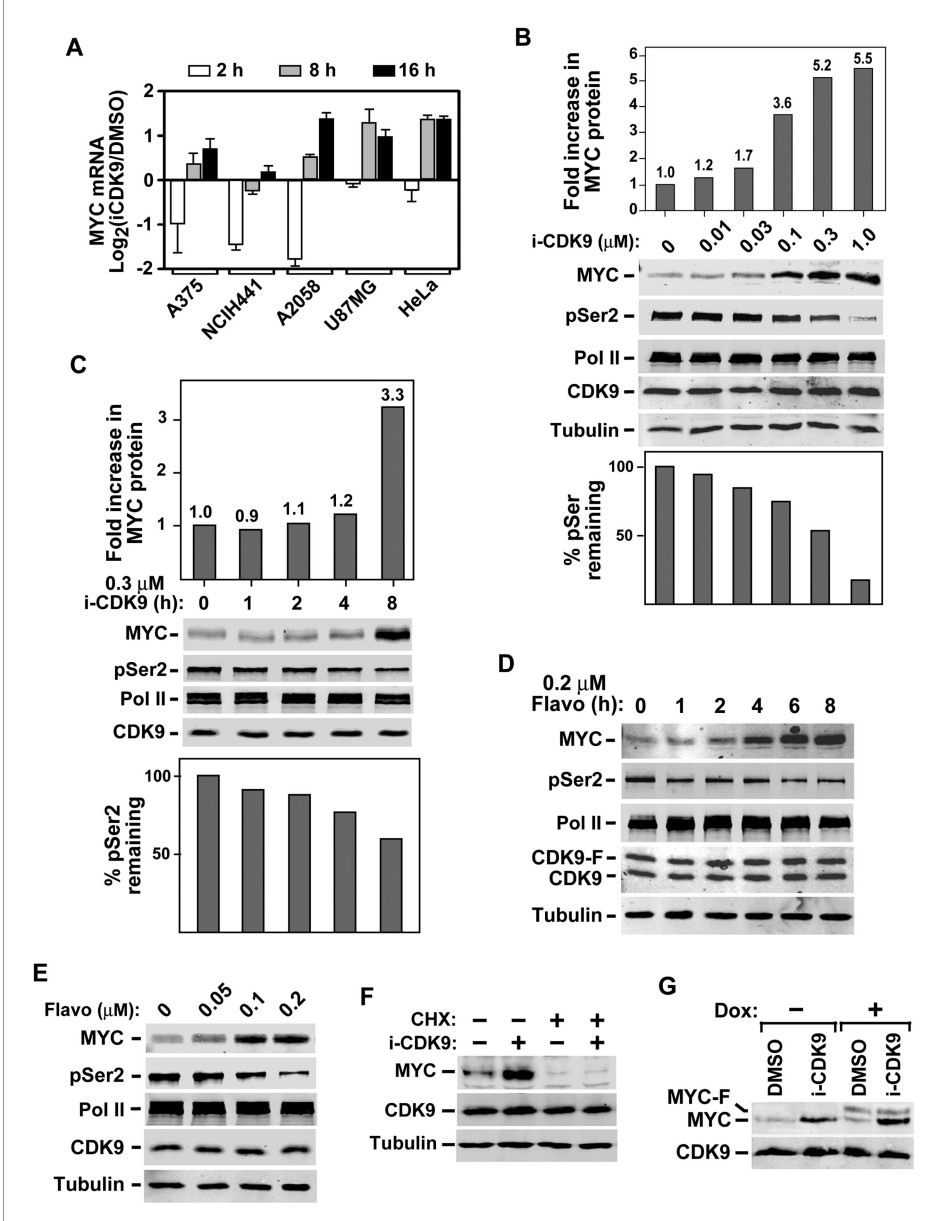

**Figure 3**. Induction of MYC mRNA production in response to sustained inhibition of CDK9 by i-CDK9 and the requirement of MYC's natural genomic structure in this process. (**A**) The indicated tumor cell lines were treated with i-CDK9 (0.5 µM) for 2–16 hr and the MYC mRNA levels, which were divided by those in the DMSO-treated cells and averaged from three independent replicates of DNA microarray analysis, were shown as log2 values. (**B**) HeLa cells were treated with the indicated concentrations of i-CDK9 for 8 hr and the various proteins in the total cell lysates were detected by immunoblotting as indicated. Quantifications of the levels of MYC protein and Ser2-phosphoryalted Pol II (pSer2) in the lysates were shown above and below the immunoblots, respectively. (**C**) HeLa cells were treated with 0.3 µM i-CDK9 for the indicated number of hr and cell lysates were obtained and analyzed as in **B**. (**D** and **E**) HeLa cells were treated with either 0.3 µM flavopiridol for the indicated time periods (**D**) or 8 hr with the indicated concentrations of flavopiridol (**E**) and analyzed by immunoblotting as in **B**. (**F**) Cells were pretreated with (+) or without (−) cycloheximide (CHX) prior to incubation with i-CDK9. The levels of the indicated proteins were examined by immunoblotting. (**G**) A HeLa-based cell line containing the stably transfected, doxycycline (Dox)-inducible MYC-F-expressing plasmid driven by the CMV promoter was pretreated with (+) or without (−) Dox prior to incubation with i-CDK9 or DMSO. MYC and MYC-F expressed from the endogenous *MYC* locus and the transfected plasmid, respectively, were detected by immunoblotting.

*Figure 3. continued on next page*

*Figure 3. Continued*

The following figure supplements are available for figure 3:

**Figure supplement 1**. Biphasic response of MYC mRNA production throughout the course of CDK9 inhibition by i-CDK9.

**Figure supplement 2**. HEXIM1 expression is continuously suppressed throughout the entire course of i-CDK9 treatment of five different tumor cell lines.

separate time-course analysis, the 8-hr incubation with i-CDK9 (0.3 μM) induced the MYC level by 3.3-fold, whereas the same condition reduced the pSer2 level by only 40% (*Figure 3C*). Thus, MYC expression was highly responsive to i-CDK9 treatment and occurred at a time point earlier and drug concentration lower than those required to achieve a significant suppression of Ser2 phosphorylation on the Pol II CTD.

Notably, a very similar MYC induction pattern was also produced by the pan-CDK inhibitor flavopiridol. Just like i-CDK9, flavopiridol also significantly elevated the MYC protein level and this effect required a shorter exposure time (e.g., 4 hr) and lower drug concentration (e.g., 0.1 μM) than those needed to fully reduce the global pSer2 level (*Figure 3D,E*). The nearly identical activities displayed by i-CDK9 and flavopiridol, which are very different in their structures, suggest that the inhibition of CDK9 but not some other unknown enzymes was the cause of *MYC* induction.

## i-CDK9 enhances the MYC mRNA production but not protein stability

To determine whether the induction of *MYC* expression by i-CDK9 requires continuous protein synthesis, we treated cells with or without the protein synthesis inhibitor cycloheximide (CHX) prior to incubation with i-CDK9. Without the pre-treatment, i-CDK9 significantly increased the MYC protein level as expected (*Figure 3F*, lanes 1–2). In the presence of CHX, however, MYC production was essentially wiped out in both the i-CDK9-treated and i-CDK9-untreated cells (lanes 3 and 4). This result, together with the above demonstration that sustained i-CDK9 treatment dramatically increased the MYC mRNA level (*Figure 3A*), indicates that MYC is a labile protein and that its prominent accumulation in i-CDK9-treated cells was most likely due to its enhanced mRNA production but not protein stability.

Interestingly, although i-CDK9 efficiently induced expression from the endogenous *MYC* locus, it failed to activate production of the Flag-tagged MYC (MYC-F) from a stably transfected, doxycycline-inducible expression construct driven by the CMV promoter (*Figure 3G*), indicating that i-CDK9 cannot activate MYC transcription from this heterologous promoter.

## Efficient i-CDK9-induced transfer of P-TEFb from 7SK snRNP to BRD4 correlates with induction of MYC transcription

A number of stress-inducing agents/conditions are known to cause the release of P-TEFb from 7SK snRNP, which serves as the principal reservoir of uncommitted P-TEFb activity in the nucleus (reviewed in *Zhou et al., 2012*). The released P-TEFb then joins the bromodomain protein BRD4 to form the BRD4-P-TEFb complex for activation of many PRGs (reviewed in *Zhou et al., 2012*). To determine whether i-CDK9 could also induce the transfer of P-TEFb from 7SK snRNP to BRD4, we examined the interactions of immunoprecipitated Flag-tagged CDK9 (CDK9-F) with HEXIM1, a signature component of 7SK snRNP, and BRD4 by Western blotting (*Figure 4A*). CDK9-F was stably expressed in the HeLa-based F1C2 cells (*Yang et al., 2001*) that were treated with a wide range of i-CDK9 concentrations for 8 hr.

Indeed, i-CDK9 gradually decreased the amounts of HEXIM1 bound to CDK9-F and at the same time increased the BRD4-CDK9-F binding in a dosage-dependent manner (*Figure 4A*). Just like the induction of MYC expression (*Figure 3B*), the disruption of 7SK snRNP and formation of BRD4-P-TEFb were highly sensitive to i-CDK9 and required as little as 0.1 μM of the drug (*Figure 4A*, compare lane 8 to lane 6). At this concentration, i-CDK9 only slightly decreased the global pSer2 level (*Figure 4A*, compare between lanes 1 and 3) but began to markedly induce *MYC* expression (*Figure 3B*). In a time course analysis, i-CDK9 (0.3 μM)-induced disruption of 7SK snRNP at a time point (1 hr) that was much

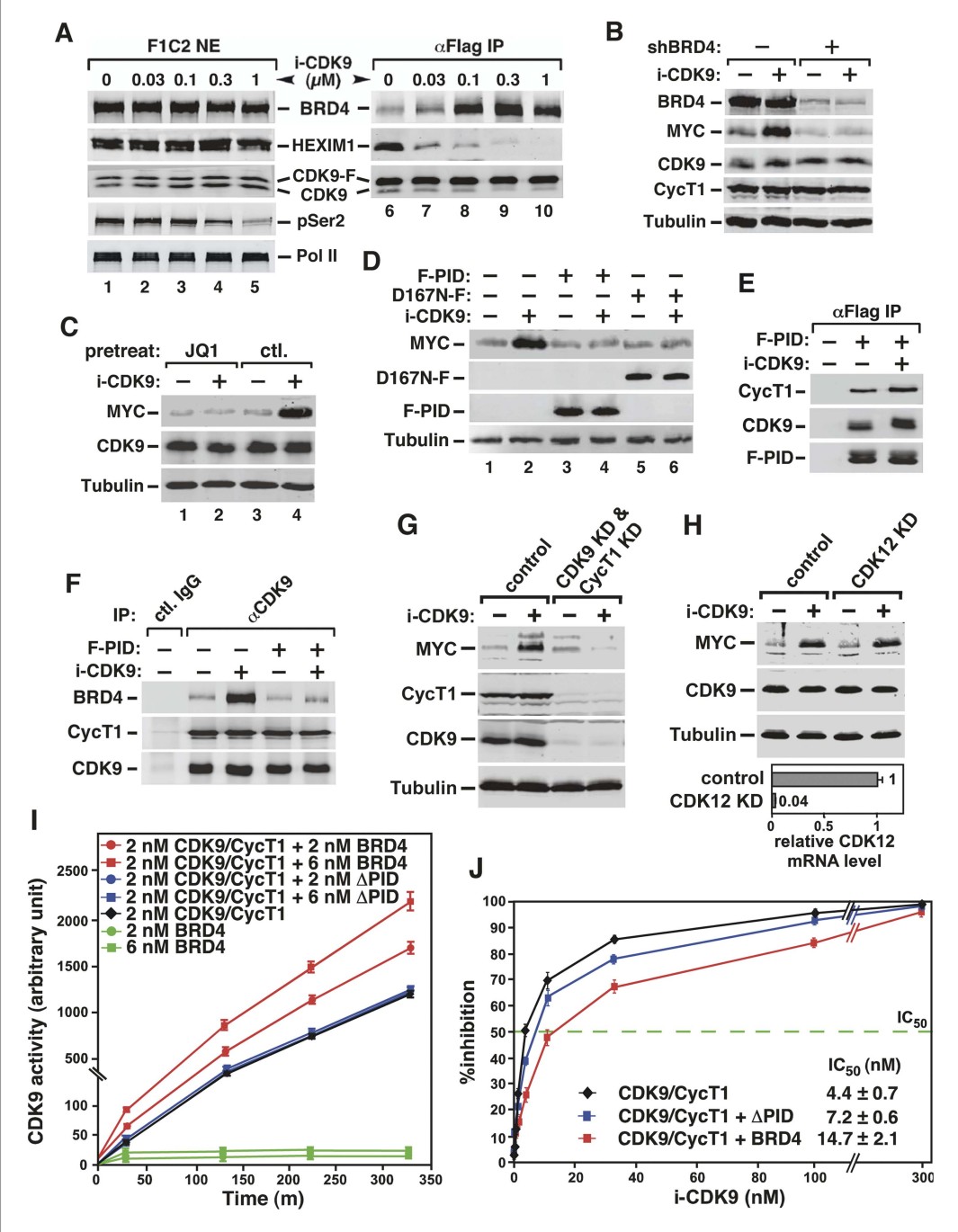

**Figure 4.** Activation of *MYC* transcription by i-CDK9 depends on induced transfer of kinase-active P-TEFb from 7SK snRNP to BRD4, binding of the BRD4-P-TEFb complex to acetylated MYC chromatin template, and BRD4-mediated increase in CDK9's catalytic activity and resistance to inhibition. (**A**) The HeLa-based F1C2 cells stably expressing CDK9-F were incubated with the indicated concentrations of i-CDK9. Nuclear extracts (NE) and the anti-CDK9-F immunoprecipitates (IP) derived from NE were analyzed by immunoblotting to detect the indicated proteins. (**B** and **C**) Lysates of HeLa cells expressing the BRD4-specific shRNA (shBRD4; **B**) or pretreated with JQ1 or the control enantiomer (ctl.; **C**) were incubated with (+) or without (−) i-CDK9 and analyzed by immunoblotting for the indicated proteins. (**D–H**) Cells were first transfected with plasmids expressing F-PID (**D**, **E** and **F**), D167N-F (**D**), or shCycT1 (**G**) or transfected with siRNAs specific for CDK9 (**G**) or CDK12 (**H**) and then treated with i-CDK9 or DMSO (−). NE (**D**, **G** and **H**) and immunoprecipitates (IP) obtained from NE with anti-Flag mAb (**E**) or anti-CDK9 antibodies or rabbit total IgG (**F**) were examined by immunoblotting for the indicated proteins. The relative CDK12 mRNA

*Figure 4. continued on next page*

*Figure 4. Continued*

levels were analyzed by qRT-PCR at the bottom in **H**, with the level in cells transfected with a control siRNA set to 1. (**I** and **J**) In vitro kinase reactions containing a synthetic Pol II CTD peptide (CDK7tide) as the substrate and CDK9-CycT1 (Invitrogen) as the kinase were conducted in the presence of the indicated amounts of WT or ΔPID BRD4. Phosphorylation of the peptide was measured over the indicated time periods and plotted in **I**, with the error bars representing mean ± SD from three independent experiments. The indicated amounts of i-CDK9 were added to the reactions in **J**, and its inhibition of CDK9 phosphorylation of the peptide was measured and plotted with the inhibitory IC50 shown.

The following figure supplements are available for figure 4:

**Figure supplement 1**. i-CDK9 (0.3 μM) induces disruption of 7SK snRNP at a time point much earlier than that required to cause about 50% reduction in global pSer2.

**Figure supplement 2**. JQ1 decreases associations of both BRD4 and CDK9 with the *MYC* locus.

**Figure supplement 3**. JQ1 blocks the i-CDK9-induced MYC expression in H1792 and A2058 cells.

**Figure supplement 4**. MYC induction by 0.3 μM i-CDK9 can be subsequently shut off by 2 μM of the drug.

**Figure supplement 5**. Examination of the purity and concentrations of WT and ΔPID BRD4 used in the CDK9 kinase assay.

---

earlier than that required to cause about 50% reduction in global pSer2 (8 hr; *Figure 4—figure supplement 1*). Together, these results reveal the remarkable efficiency with which i-CDK9 induces the transfer of P-TEFb from 7SK snRNP to BRD4.

## i-CDK9 induction of MYC expression requires BRD4 and its association with acetylated chromatin

Given the above demonstration that the i-CDK-induced MYC transcription occurred under the same conditions that stimulated the transfer of P-TEFb from 7SK snRNP to BRD4, it is important to confirm that BRD4 is indeed required for the MYC induction. Toward this goal, BRD4 knockdown (KD) was performed with a specific shRNA (shBRD4), whose expression was induced by Cre recombinase (*Yang et al., 2008*). The loss of BRD4, verified by Western blotting, dramatically reduced both basal and the i-CDK9-induced MYC protein production (*Figure 4B*).

To determine if the binding of BRD4 to acetylated chromatin is required for i-CDK9 to induce *MYC* expression, we tested whether the BET bromodomain inhibitor JQ1 could block the induction. JQ1 is known to competitively bind to the acetyl-lysine recognition pocket within BRD4's bromodomains, leading to the dissociation of the BRD4-P-TEFb complex from acetylated chromatin (*Zuber et al., 2011*; *Li et al., 2013*). Consistent with the results obtained in other cell types (*Mertz et al., 2011*; *Zuber et al., 2011*), JQ1 decreased the associations of both BRD4 and CDK9 with the *MYC* locus in HeLa cells (*Figure 4—figure supplement 2*). More importantly, JQ1 also completely abolished the MYC induction by i-CDK9 in HeLa (*Figure 4C*) as well as the lung cancer cell line H1792 and the melanoma cell line A2058 (*Figure 4—figure supplement 3*), whereas the control enantiomer was ineffective in this regard. These observations, in conjunction with the shBRD4 result above, confirm that the interaction between BRD4 and acetylated chromatin is required for i-CDK9 to induce MYC expression in diverse cell types.

## The kinase-active P-TEFb and its interaction with BRD4 are required for i-CDK9 to induce MYC expression

We next asked whether the enhanced interaction between BRD4 and P-TEFb as a result of i-CDK9 treatment (*Figure 4A*) is also required for the *MYC* induction. To this end, the ability of i-CDK9 to induce MYC protein production was tested in the presence of the overexpressed P-TEFb-interacting domain (PID; aa1209-1362) of BRD4 (*Bisgrove et al., 2007*). Western analyses indicate that the PID efficiently suppressed the induction (*Figure 4D*, lanes 3 and 4), likely due to its own strong interaction

with CDK9 and CycT1 in i-CDK9-treated cells (*Figure 4E*), which in turn interfered with the interaction of endogenous BRD4 with P-TEFb (*Figure 4F*).

Not only was the enhanced physical interaction between P-TEFb and BRD4 essential for the i-CDK9-induced MYC expression, more importantly, the wild-type (WT) kinase activity of the BRD4-bound P-TEFb was also required. This point is illustrated by the demonstration that the overexpressed kinase-inactive CDK9 mutant D167N, which binds to BRD4-like WT CDK9 (*Yang et al., 2005*) and acts *dominant-negatively* to suppress the activity of endogenous CDK9 (*Garber et al., 2000*), prevented i-CDK9 from inducing MYC (*Figure 4D*, lanes 5 and 6).

The dependence on catalytically active P-TEFb for *MYC* induction was further demonstrated by the observation that the initial MYC induction by a low level (e.g., 0.3 µM) of i-CDK9 was subsequently shut off by a higher i-CDK9 concentration (e.g., 2 µM, see *Figure 4—figure supplement 4*). No MYC induction was detected also when 2 µM i-CDK9 was added into the medium at time zero (see Figure 7A below). Finally, the RNAi-mediated KD of cellular CDK9 and CycT1 levels also completely abolished the i-CDK9-induced MYC expression (*Figure 4G*). In contrast, KD of CDK12, another reported CTD kinase, failed to prevent i-CDK9 from inducing MYC (*Figure 4H*). Taken together, these data strongly support the notion that the i-CDK9 induction of *MYC* expression depends on the enhanced interaction of BRD4 with catalytically active P-TEFb released from 7SK snRNP as well as the binding of the BRD4-P-TEFb complex to acetylated MYC chromatin template.

## Binding of BRD4 to P-TEFb increases CDK9's catalytic activity and resistance to inhibition

The above data have revealed an apparent paradox concerning the role of CDK9 kinase during i-CDK9 induction of MYC expression. Although the induction was caused by a highly specific CDK9 inhibitor, the actual process was found to require the interaction of BRD4 with catalytically active CDK9. One possible explanation for these seemingly contradictory observations is the above demonstrations that the MYC induction was highly sensitive and occurred at a i-CDK9 concentration lower and time point earlier than those required to completely abolish the nuclear CDK9 kinase activity. In addition to this kinetic advantage displayed by the very sensitive MYC induction process, we also investigated whether the i-CDK9-induced BRD4-P-TEFb binding could also directly affect the kinase activity of CDK9.

To this end, the ability of recombinant CDK9-CycT1 (Invitrogen) to phosphorylate a synthetic Pol II CTD peptide (termed CDK7tide; Bio-Synthesis, Inc.) was measured in the presence of either WT BRD4 or a BRD4 mutant lacking the C-terminal P-TEFb-interacting domain (∆PID). The BRD4 proteins were affinity-purified from transfected HEK293T cells under highly stringent conditions to strip away their associated factors (*Figure 4—figure supplement 5*). While WT BRD4 was able to increase CDK9 kinase activity dose dependently up to 2.2-fold, ∆PID lacked this ability (*Figure 4I*), indicating that the physical interaction between BRD4 and P-TEFb was essential for the elevated CDK9 activity. It is worth pointing out that different from a recent study reporting that BRD4 is an atypical kinase that can directly phosphorylate the Pol II CTD on Ser2 (*Devaiah et al., 2012*), neither WT nor ∆PID BRD4 alone was able to cause phosphorylation of the CTD peptide in the absence of P-TEFb (*Figure 4I*).

In addition to increasing CDK9's catalytic activity, BRD4 was also found to render CDK9 less sensitive to inhibition by i-CDK9, whereas BRD4∆PID was less effective in this regard (*Figure 4J*). For example, in the absence of BRD4, it took only $4.4 \pm 0.7$ nM i-CDK9 to achieve 50% inhibition of CDK9 (i.e., $IC_{50} = 4.4 \pm 0.7$ nM). However, the addition of WT BRD4 into the reaction protected CDK9 against i-CDK9 and increased $IC_{50}$ to $14.7 \pm 2.1$ nM. BRD4∆PID, on the other hand, displayed a much weaker effect by changing the $IC_{50}$ value to $7.2 \pm 0.6$ nM (*Figure 4J*). These data indicate that the i-CDK9-mediated MYC induction benefited not only from efficient release of P-TEFb from 7SK snRNP, but the drug-enhanced BRD4-P-TEFb interaction can also directly promote CDK9's kinase activity and resistance to inhibition, in addition to its recruitment of more P-TEFb to the MYC locus.

## i-CDK9-induced MYC expression is associated with increased occupancy of P-TEFb, BRD4, Pol II, acetyl-H3/H4 and enhanced Ser2 phosphorylation on Pol II CTD at the MYC locus

To further probe the molecular basis underlying *MYC* induction by i-CDK9, we performed the classic ChIP-qPCR analysis to examine the interactions of BRD4, CDK9, total Pol II, and the Ser2-phosphoryalted

Pol II with the *MYC* locus before and after treatment with 0.3 µM i-CDK9 for 8 hr. The first noticeable major change caused by i-CDK9 is the significant increase in BRD4's occupancy across the entire MYC locus (*Figure 5A,B*), which explains why the MYC induction was highly sensitive to the BRD4 inhibitor JQ1 (*Figure 4C*) and shBRD4 (*Figure 4B*). Correlating with this increase and consistent with the i-CDK9-induced BRD4-P-TEFb interaction, there was also significantly elevated CDK9 binding to the MYC locus (*Figure 5B*). Although the elevation in CDK9 binding was mostly BRD4 dependent (*Figure 5—figure supplement 1*), the distribution pattern of CDK9 was somewhat different from that of BRD4 (*Figure 5B*). This difference could be caused by P-TEFb's dissociation from BRD4 and joining the Pol II elongation complex once it is recruited to the MYC chromatin template.

Consistent with the demonstration that the i-CDK9-induced *MYC* expression is due to increased transcription by Pol II (*Figure 3A,F*), i-CDK9 also markedly enhanced the concentration of Pol II across the entire MYC locus (*Figure 5B*). Furthermore, despite the incubation with 0.3 µM i-CDK9 for 8 hr, a condition that was shown above to decrease the global Ser2 phosphorylation on Pol II CTD by about 40% (*Figure 3B*), the pSer2 level was found to still increase at the MYC locus (*Figure 5B*). Comparing to CDK9 and total Pol II, the pSer2 distribution shifted toward the 3′ end of the gene, which is consistent with the patterns detected on many other actively transcribed genes (*Rahl et al., 2010*; *Zhou et al., 2012*). In contrast to the situation observed on the i-CDK9-induced MYC gene, on HEXIM1, a gene that was permanently inhibited by i-CDK9 (*Figure 3—figure supplement 2*), the same treatment decreased the levels of both total Pol II and Pol II with pSer2 CTD (*Figure 5—figure supplement 2*).

Finally, while the levels of acetylated histones H3 (Ac-H3) and H4 (Ac-H4) in whole cell extracts remained unchanged upon exposure to i-CDK9 (*Figure 5—figure supplement 3*), they were found to increase across the MYC locus especially at locations surrounding the major P1 promoter (*Figure 5C*). Although the underlying mechanism for this increase is unclear, it explains well the drug-promoted BRD4 binding to the MYC locus. Interestingly, even at early time points of i-CDK9 treatment (0, 1 and 2 hr) when MYC transcription was yet to be induced, the acetylation state of the MYC promoter was already different from that of the HEXIM1 promoter. As shown in *Figure 5D*, the levels of both Ac-H3 and Ac-H4 at the MYC promoter began to increase at this early stage, with the more robust increase observed for Ac-H4. In contrast, at the HEXIM1 promoter, i-CDK9 caused a drastic decrease in the Ac-H3 level but a small increase of Ac-H4. As for the BRD4 level, it displayed a marked decrease at the MYC promoter at 1 hr post i-CDK9 treatment, but began to rebound by 2 hr. At the HEXIM1 promoter, however, it showed a sustained reduction throughout the entire period (*Figure 5D*). It is likely that these early differences at the chromatin level predispose a transiently repressed gene such as MYC and a permanently repressed one such as HEXIM1 to completely different expression states later on during the i-CDK9 treatment.

Taken together, the data presented thus far indicate that the i-CDK9-induced MYC expression is likely caused by a combination of events: (1) the induced transfer of P-TEFb from 7SK snRNP to BRD4; (2) BRD4's promotion of CDK9's kinase activity and resistance to inhibition; and (3) favorable chromatin changes at an early stage of the i-CDK9 treatment that lead to subsequent increase in the occupancy of the BRD4-P-TEFb complex at the MYC locus, promotion of Ser2 phosphorylation on the Pol II CTD, and finally productive transcriptional elongation.

## i-CDK9 affects expression of other BRD4-dependent primary response genes similarly as it does for MYC

MYC is a classic example of the so-called PRGs, which are a set of genes that can be induced in response to both extracellular and intracellular signals without the requirement for de novo protein synthesis (*Fowler et al., 2011*). Given the above demonstrations that BRD4 and its interaction with P-TEFb played an essential role in i-CDK9-induced MYC expression, we next investigated whether other BRD4-dependent primary response (BDPR) genes such as those identified in bone marrow-derived macrophages (*Figure 6A*; [*Hargreaves et al., 2009*]) might behave similarly as MYC in their response to i-CDK9. To this end, we performed gene set enrichment analysis (GSEA) of expression microarray data obtained from HeLa cells treated with either DMSO or i-CDK9 for 2 and 8 hr (*Figure 6B,C*). At the 2-hr time point, GSEA reveals a marked i-CDK9-induced reduction of expression (normalized enrichment score or NES = −2.28) for essentially all the 27 genes within the gene set (*Figure 6A,B*). However, at the 8-hr time point, the trend was completely reversed with virtually all the

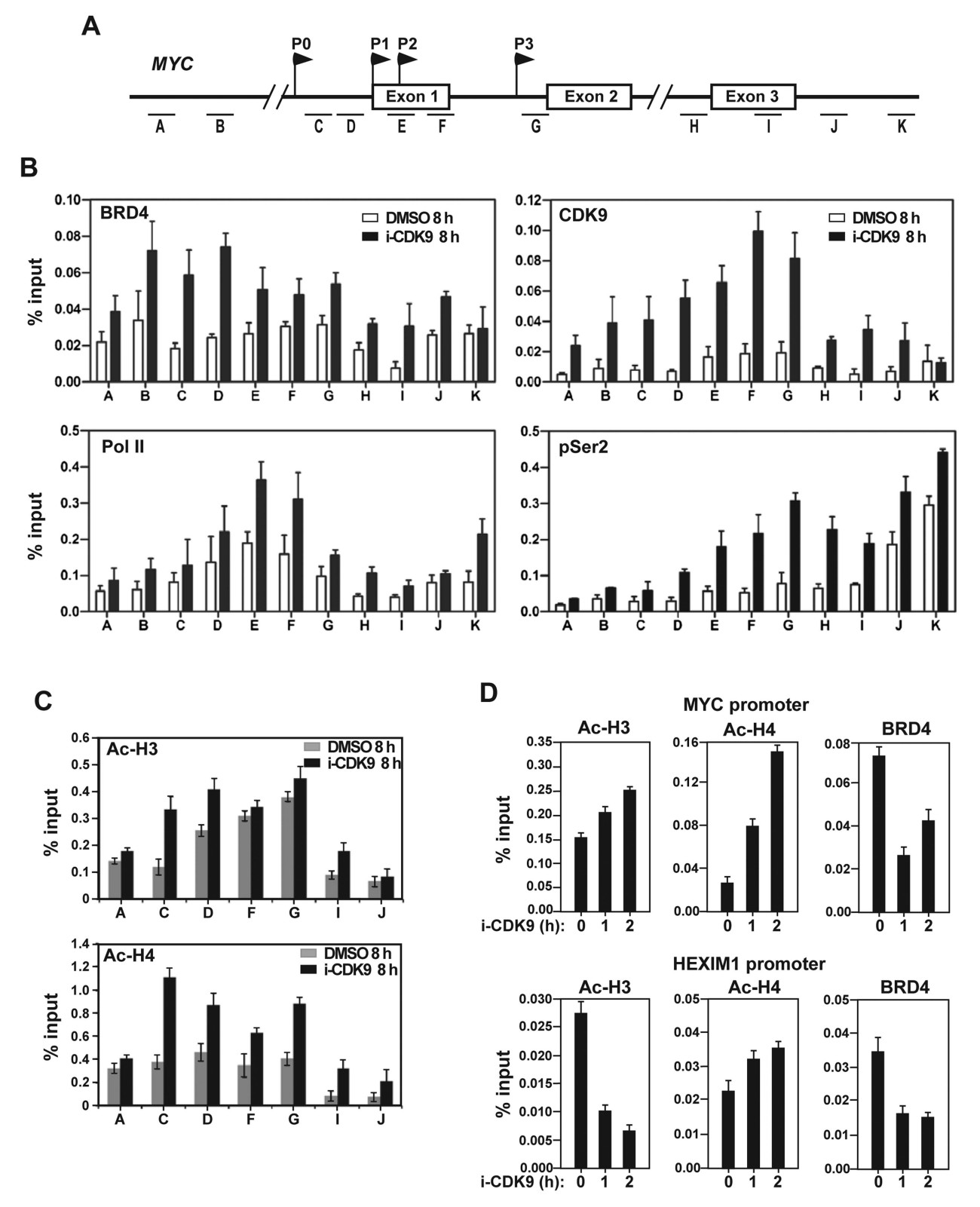

**Figure 5**. Treatment with i-CDK9 (0.3 $\mu$M for 8 hr) increases the levels of P-TEFb, BRD4, total Pol II, Pol II with pSer2 CTD and acetyl-H3/H4 at the MYC locus. (**A**) Genomic structure of the MYC locus. Arrows indicate the positions and direction of the four MYC promoters P0 to P3. The small horizontal bars labeled with letters A to K mark the positions of 11 amplicons generated by quantitative PCR (qPCR) analysis of the ChIP DNA. (**B** and **C**) HeLa cells were

*Figure 5. continued on next page*

*Figure 5. Continued*

treated with either i-CDK9 or DMSO and subjected to ChIP-qPCR analysis to determine the levels of the indicated factors bound to the MYC locus. The signals were normalized to those of input; and the error bars in all panels represent mean ± SD from three independent experiments. (**D**) HeLa cells were treated with 0.3 μM i-CDK9 for the indicated time periods and subjected to ChIP-qPCR analysis to determine the levels of the indicated factors bound to the MYC locus at position C and the HEXIM1 locus at position L (see *Figure 5—figure supplement 2*). The signals were normalized to those of input; and the error bars represent mean ± SD from three independent experiments.

The following figure supplements are available for figure 5:

**Figure supplement 1**. The i-CDK9-induced increase in CDK9's binding to the MYC locus is mostly BRD4-dependent.

**Figure supplement 2**. Treatment with i-CDK9 (0.3 μM for 8 hr) decreases the levels of both total Pol II and Pol II with pSer2 CTD at the HEXIM1 locus.

**Figure supplement 3**. i-CDK9 does not affect the cellular levels of acetylated histones H3 and H4.

genes displaying significant up-regulation by i-CDK9 (*Figure 6C*; NES = +2.26). Thus, the biphasic transcriptional signature of the curated BDPR genes is the same as that of MYC in their response to i-CDK9.

## i-CDK9 has little effect on TR of BDPR genes

Another interesting observation about the 23 BDPR genes that had detectable Pol II ChIP-seq signals in HeLa cells (*Figure 6A*, bold face type) is that they showed little change in their Pol II TR values after the exposure to i-CDK9 for 8 hr (*Figure 6D*). In contrast, a significant increase in TR was found for most of the remaining (*Figure 6D* upper panel) or 23 randomly selected genes (lower panel) in the ChIP-seq database. A close examination of the distribution patterns of Pol II on three well-studied PRGs, MYC, FOS, and JUNB, which happen to be highly important proto-oncogenes, reveals that i-CDK9 caused a significant and fairly uniform increase in Pol II occupancy across the entire length of these genes (*Figure 6E–G*). Notably, the distribution of Pol II on MYC as revealed by ChIP-seq is remarkably similar to that obtained by the classic ChIP-qPCR (*Figure 5B*). These results are not only consistent with the induced expression of these genes by i-CDK9, but also explain why their TR values underwent little change in the process. Taken in aggregate, the data so far strongly support the notion that the induced transfer of P-TEFb from 7SK snRNP to BRD4 and the BRD4-mediated protection and enhancement of P-TEFb's activity and recruitment to chromatin templates are the primary driving force behind the induction of MYC and other BDPR genes by i-CDK9.

## MYC facilitates P-TEFb phosphorylation of Pol II CTD and increases binding to BRD4-P-TEFb upon CDK9 inhibition

Having identified the mechanism of MYC induction by i-CDK9, we also wanted to determine the biological significance of this phenomenon. Given that MYC induction was triggered by inhibition of cellular CDK9 kinase, we hypothesized that MYC may normally facilitate CDK9's function and that the elevated MYC expression in i-CDK9-treated cells is therefore a cellular attempt to compensate for the loss of CDK9. To test this hypothesis, we first determined whether the shRNA-mediated MYC KD would affect the global pSer2 levels before and after the treatment with i-CDK9. Indeed, even though the KD was incomplete, it markedly reduced pSer2 levels in both untreated cells as well as in cells exposed to a range of i-CDK9 concentrations (*Figure 7A*), revealing a requirement of MYC for CDK9 to phosphorylate the Pol II CTD under both normal and inhibitory conditions.

Not only did the KD of MYC expression negatively affect the ability of CDK9 to phosphorylate Ser2, the treatment of cells with either JQ1, the BET bromodomain inhibitor that suppressed the BRD4-dependent MYC expression (*Figure 4C*), or 10,058-F4, a small molecule inhibitor that inhibits the MYC-MAX interaction to prevent transactivation of genes targeted by this heterodimer (*Rahl et al., 2010*), also slightly decreased the cellular pSer2 levels (*Figure 7B*, lane 4; *Figure 7C*, lane 2). More importantly, when JQ1 or 10,058-F4 was used in combination with i-CDK9, global pSer2 dropped to levels that were much lower than those caused by i-CDK9 alone (*Figure 7B*, lane 3; *Figure 7C*, lane 4). All together, these results are consistent with the idea that MYC normally functions

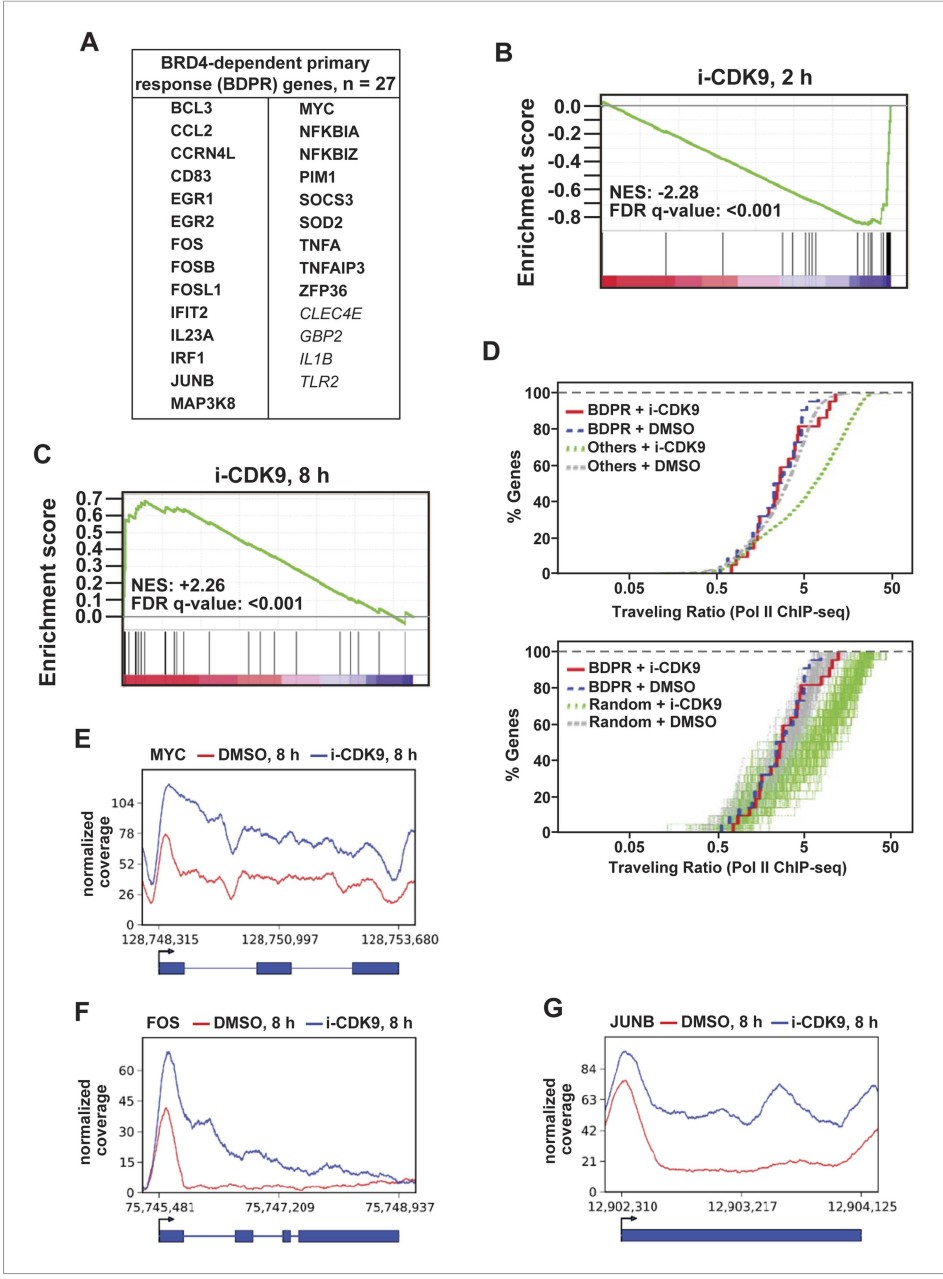

**Figure 6**. i-CDK9 affects the expression of other BRD4-dependent primary response genes similarly as it does to MYC. (**A**) The list of 27 curated BRD4-dependent primary response (BDPR) genes identified in bone marrow-derived macrophages is displayed in alphabetical order. The 23 genes in bold face type had detectable Pol II signals in HeLa cells as revealed by ChIP-seq analysis. (**B** and **C**) GSEA results for the 27 BDPR genes at 2 hr (**B**) and 8 hr (**C**) post CDK9 inhibition. NES: Normalized Enrichment Score; FDR: False Discovery Rate. (**D**) Distribution of Pol II-bound genes with a given TR as determined by ChIP-seq. The genes are grouped by the indicated gene types and treatment conditions. The top panel compares the 23 BDPR genes to the remaining Pol II-bound genes in the genome, and the bottom compares the BDPR genes to 23 randomly selected genes. (**E, F, G**) Occupancy of Pol II across three representative BDPR genes as revealed by ChIP-seq. The read coverage is shown for the entire gene plus a margin on either side equal to 7% of the gene length.

to facilitate P-TEFb's phosphorylation of the Pol II CTD on Ser2 and that decreasing the expression or activity of MYC eliminates this beneficial effect.

The observation that MYC can bind and recruit P-TEFb to numerous MYC-target genes to activate transcription (*Kanazawa et al., 2003*; *Rahl et al., 2010*) provides a plausible explanation for its

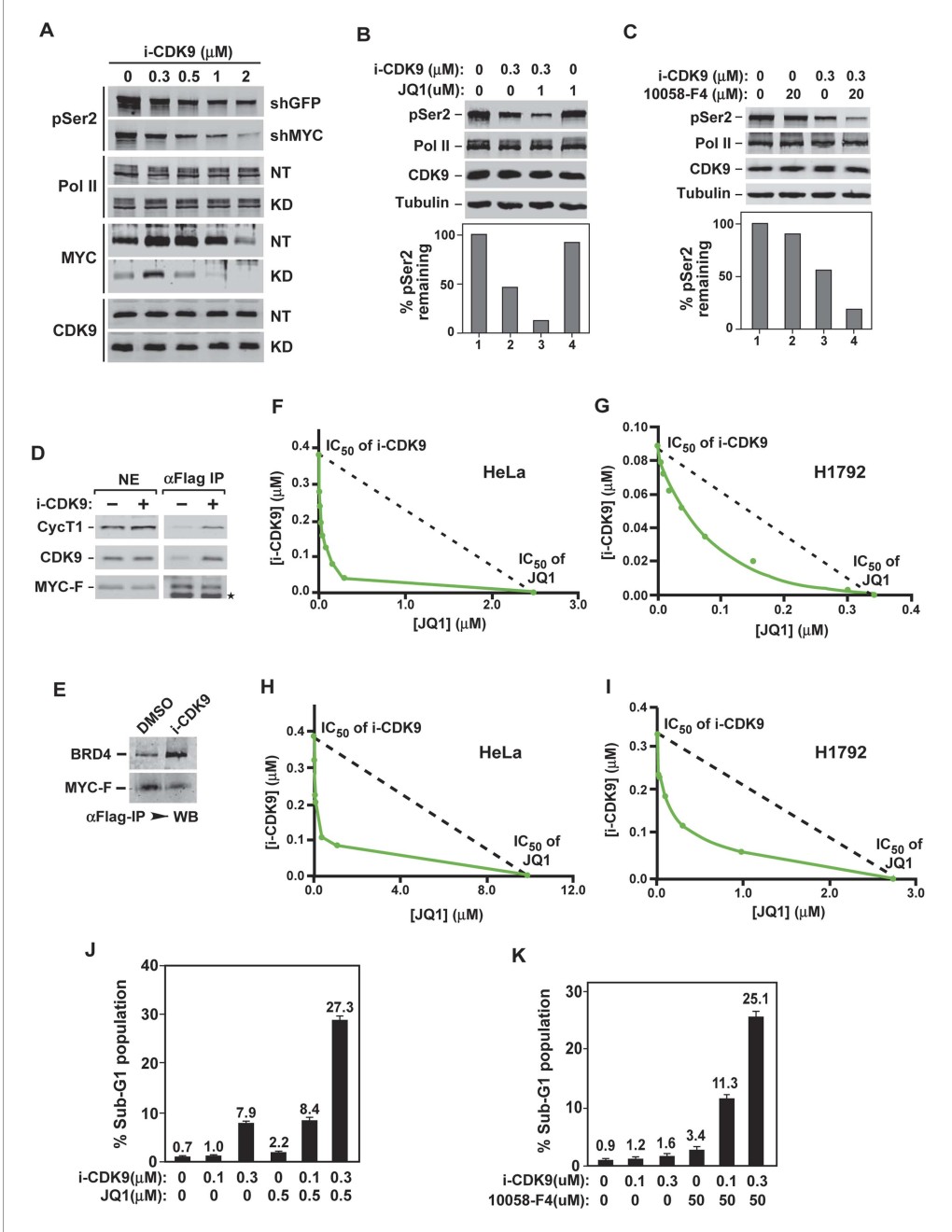

**Figure 7**. Simultaneous inhibition of CDK9 and MYC synergistically induces growth arrest and apoptosis of cancer cells due to the fact that MYC facilitates P-TEFb phosphorylation of Pol II CTD and increases binding to BRD4-P-TEFb upon CDK9 inhibition. (**A**) Lysates of HeLa cells expressing the indicated shRNA and exposed to increasing concentrations of i-CDK9 were analyzed by immunoblotting for the indicated proteins. (**B** and **C**) Lysates of HeLa cells treated with the indicated drugs and their concentrations were analyzed by immunoblotting, with quantification of the pSer2 signals shown at the bottom. (**D** and **E**) Nuclear extracts (NE) of HeLa-based cells expressing MYC-F and untreated (−) or treated with i-CDK9 or DMSO were subjected to anti-Flag immunoprecipitation. The immunoprecipitates (IP) were examined by immunoblotting for the indicated proteins. (**F**, **G**, **H**, and **I**) HeLa (**F** and **H**) and H1792 (**G** and **I**) cells were incubated with JQ1 or i-CDK9 alone or together at various concentrations. The concentrations of each drug (IC$_{50}$), either used as a single agent or in combination, that caused 50% of cells to show growth inhibition in Celltiter-Glo assay (**F** and **G**) or produce Caspase 3/7 (**H** and **I**) were plotted using the isobologram method. The dotted lines denote the IC$_{50}$ values of i-CDK9 and JQ1 had the effects of the two compounds been simply additive. (**J** and **K**) HeLa cells were treated with the indicated concentrations of i-CDK9 plus

*Figure 7. continued on next page*

Figure 7. Continued

JQ1 (**J**) or i-CDK9 plus 10,058-F4 (**K**) and measured by flow cytometry for propidium iodide (PI)-stained sub-G1 population. The error bars represent mean ± SD from three independent measurements.

promotion of P-TEFb's pSer2. In addition to enhancing MYC's expression, we also investigated whether i-CDK9 may also directly affect the interaction of MYC with BRD4-P-TEFb. Indeed, the amounts of CDK9, CycT1, and BRD4 bound to the immunoprecipitated MYC-F were found to increase upon the exposure to i-CDK9 (*Figure 7D,E*), revealing a dual mechanism used by cells to compensate for the loss of CDK9 activity by targeting the BRD4-P-TEFb complex to essential genes via a MYC-dependent pathway.

## Simultaneous inhibition of CDK9 and MYC synergistically induces growth arrest and apoptosis of cancer cells

Because the loss of CDK9 activity in i-CDK9-treated cells can be compensated by an increase in MYC's expression and interaction with P-TEFb, we reasoned that any meaningful antitumor effect caused by inhibiting CDK9 must involve the simultaneous suppression of the BRD4-dependent MYC expression. To test this hypothesis, we first performed cell proliferation assays and used the isobologram method to determine whether the combined effects of i-CDK9 and JQ1 were additive, synergistic, or antagonistic (*Tallarida, 2006*). Data matrices were generated by measuring anti-proliferative effects (IC$_{50}$) of i-CDK9 and JQ1 as a single agent or in combination on both HeLa and H1792, a non-small cell lung cancer cell line with significant MYC amplification (CN = 8). Our data clearly show that the combination of i-CDK9 and JQ1 synergistically inhibited growth of both cell lines (*Figure 7F,G*).

In addition to inhibition of cell growth, the combination of i-CDK9 and JQ1 also synergistically induced apoptosis of both HeLa and H1792 cells as demonstrated by isobolograms that measure the production of Caspase 3/7 in treated cells (*Figure 7H,I*). Furthermore, strong and synergistic apoptosis caused by i-CDK9 and JQ1 was also observed by flow cytometry measurement of propidium iodide (PI)-stained sub-G1 population of HeLa cells that contained fragmented DNA (*Figure 7J*). Finally, in addition of JQ1, the MYC inhibitor 10,058-F4 was also found to synergize with i-CDK9 to induce potent cell death (*Figure 7K*). Taken all together, the data are consistent with the notion that simultaneous inhibition of CDK9's kinase activity and MYC's expression or function leads to synergistic induction of growth arrest and apoptosis of cancer cells.

## Discussion

The aberrant activation of cyclin-dependent kinases (CDKs) and dysregulation of cell cycle progression is a hallmark of many human diseases that include cancer, cardiac myopathies, and inflammatory processes. Because of this, manipulation of cell cycle progression by means of small molecule inhibitors has long been suggested as a therapeutic avenue to cure these diseases especially cancer. There are now more than 50 pharmacological CDK inhibitors that have been described (*Knockaert et al., 2002*; *Fisher, 2010*). Among these, flavopiridol has been promoted as a potent inhibitor of CDKs with a notable bias for CDK9 (*Senderowicz and Sausville, 2000*; *Chao and Price, 2001*; *Blagosklonny, 2004*). Treatment with flavopiridol has been shown to block cell cycle progression, promote differentiation, and induce apoptosis in various types of cancerous cells (*Senderowicz and Sausville, 2000*; *Blagosklonny, 2004*). However, as indicated in the current study, flavopiridol displayed only 2.9 to 28.6-fold lower inhibitory activity toward other CDK-cyclin pairs compared to CDK9-CycT1 (*Figure 1B*). This relatively weak selectivity for CDK9, coupled with the demonstrations that flavopiridol also binds strongly to duplex DNA (*Bible et al., 2000*) and many other non-CDK kinases (http://www.discoverx.com/ReferenceTreeImages/Flavopiridol.htm), raise the possibility that some of the effects attributed to this compound may not be caused by the inhibition of CDK9. Because simultaneous inhibition of both CDK9 and cell cycle CDKs makes the data interpretation difficult and confusing, a highly selective CDK9 inhibitor that can specifically block the P-TEFb-dependent elongation phase of Pol II transcription is thus very desirable and urgently needed. In this regard, the current development and characterization of i-CDK9, which demonstrated far superior selectivity against CDK9 than did flavopiridol, is both timely and valuable for our efforts to further investigate CDK9 biology as well as the therapeutic potential of P-TEFb inhibition.

One key finding from the present study is that long-term inhibition of P-TEFb by i-CDK9 potently up-regulated the expression of a number of PRGs that include MYC, FOS, JUNB, ERG1, and others. Notably, a subset of the same PRGs (FOS, JUNB, EGR1, and GADD45B) has also been shown to be potently down-regulated before they were significantly up-regulated following the treatment with the pan-CDK inhibitor flavopiridol (*Keskin et al., 2012*). As for MYC, although several mechanisms have been indicated as responsible for its elevated expression, including gene amplification, chromosomal translocation, and alteration of protein stability (*Meyer and Penn, 2008*), the BRD4-dependent mechanism identified here and used by cells to up-regulate MYC and other PRGs in response to CDK9 inhibition represents a previously uncharacterized alternative method that appears to be equally effective in promoting MYC expression.

In addition to serving as an epigenetic reader and recruitment agent to deliver active P-TEFb released from 7SK snRNP to acetylated chromatin at the MYC locus, BRD4 is also shown in the present study to use its C-terminal P-TEFb-interaction domain (PID) to directly increase the catalytic activity of CDK9 despite the fact that itself failed to display any kinase activity toward the Pol II CTD. This result is consistent with the observations reported recently by the Geyer laboratory, which employed both the synthetic CTD peptides as well as the full-length human CTD containing all 52 repeats in the kinase assays (*Itzen et al., 2014*). Likely employing a similar mechanism for activating CDK9, BRD4 is further shown in our study to also render CDK9 more resistant to inhibition when bound to P-TEFb (*Figure 4J*).

At this moment, the precise mechanism used by BRD4 PID to promote CDK9's kinase activity and protect against i-CDK9 is yet to be determined. It is interesting to note that the HIV-1 Tat protein appears to possess a similar ability to activate CDK9 upon binding to the isolated, recombinant P-TEFb (*Garber et al., 2000*). While Tat is known to bind mostly to the surface of CycT1 (*Schulze-Gahmen et al., 2014*), the PID appears to contact both CycT1 and CDK9 (*Yang et al., 2005*; *Itzen et al., 2014*). Thus, Tat and BRD4 PID may not contact P-TEFb in exactly the same manner. Despite this possible difference, the fact that the PID exhibits a sequence motif composition that is similar to that of Tat, and moreover, the conserved Tat-like motifs are also required for P-TEFb activation (*Itzen et al., 2014*), suggests that the interaction of P-TEFb with either Tat or BRD4 PID may induce a similar conformational change that is beneficial for CDK9's interaction with substrates.

The data presented in the present study support the model that the i-CDK9-induced MYC expression and interaction with P-TEFb is important to compensate for the loss of P-TEFb activity in treated cells. Given the fact that P-TEFb is a general transcription factor (*Zhou et al., 2012*) and that MYC is traditionally viewed as a sequence-specific transcription factor existing in a heterodimer with MAX (*Amati et al., 1993*), how is the elevated MYC level and binding to P-TEFb able to sustain the expression of a vast array of cellular genes that are normally dependent on P-TEFb for proper transcription (the current study and *Chao and Price, 2001*; *Shim et al., 2002*)? The answer to this question has recently come from studies showing that MYC is a transcriptional factor much more promiscuous in terms of the number and types of its target genes than what is suggested by the simple distribution and abundance of the MYC cognate sequence called the E-box in the genome.

For example, when investigating the MYC-induced global amplification of transcription in lymphocytes and ES cells, the Levens group noticed that the activation of genes that contain the E-box, which are traditionally defined as the MYC-target genes, makes up only a modest fraction of the total number of genes activated in these cells (*Nie et al., 2012*). Consistent with the idea that sequence-specific DNA binding is largely dispensable for most of MYC's transactivation function, the Price laboratory reported that the positions occupied by the MYC-MAX dimer across the human genome correlate with the Pol II transcription machinery rather than E-box elements (*Pufall and Kaplan, 2013*). Furthermore, a significant percentage of MYC-MAX is located slightly upstream of nearly all the promoter-proximally paused Pol II. These results, together with the previous demonstrations that MYC can recruit P-TEFb to its target genes to activate Pol II transcriptional elongation (*Kanazawa et al., 2003*; *Rahl et al., 2010*), implicate a critical role for this transcription factor in globally controlling the release of Pol II from pausing. This general but hitherto unknown function of MYC also explains well our present observations that blocking MYC's expression by shMYC or JQ1 or inhibiting the MYC-MAX function by 10,058-F4 further exacerbated the i-CDK9-induced global decrease in Pol II CTD phosphorylation on Ser2 (*Figure 7A–C*).

While the inhibition of CDK9 leads to elevated MYC expression and binding to P-TEFb in order to mitigate the impact of loss of cellular CDK9 activity, previous reports have shown that treating cells

with JQ1, which blocks the BRD4-dependent MYC expression, causes the release and activation of P-TEFb from 7SK snRNP (*Bartholomeeusen et al., 2012*; *Li et al., 2013*). These observations suggest the existence of a complementary and compensatory relationship between P-TEFb and MYC, and that the two transcription factors are likely part of a redundant mechanism used by cells to ensure optimal expression of key jointly controlled genes. Consistent with this idea, we show in the present study that simultaneous inhibition of CDK9's kinase activity and MYC's expression or function synergistically induced growth arrest and apoptosis of cancer cells. The importance of this result lies in its implication for future clinical applications of i-CDK9 and JQ1 or other BET bromodomain inhibitors. Since small molecule inhibitors such as i-CDK9 and JQ1, no matter how selective they are, will inevitably produce off-target effects when used at high concentrations, the demonstration that they can produce dramatic anti-tumor effects when used together at levels much lower than either alone holds the great promise of minimizing these unwanted off-target effects in future therapy.

## Materials and methods

### Experimental procedures

#### i-CDK9, antibodies, and cell lines

N2′-(trans-4-aminocyclohexyl)-5′-chloro-N6-(3-fluorobenzyl)-2,4′-bipyridine-2′,6-diamine, nicknamed i-CDK9 in the current study, was synthesized at Novartis Institutes for BioMedical Research (NIBR, Emeryville, CA).

The antibodies used in immunoblots and cell-based assays were either purchased or raised as follows: pSer2 RNA Pol II (Abcam, Cat. #ab5095), total RNA Pol II (F-12) (Santa Cruz, Cat. #sc-55492), Mcl-1 (Cell Signaling, Cat. #4572), PARP (Cell Signaling, Cat. #9542), SPT5 (Novus Biologicals, Cat. #NB110-40593), β-Actin (Sigma, Cat. #A5441), MYC (Cell Signaling, Cat. #9402), Europium-labeled anti-rabbit antibody (Perkin Elmer, Cat# AD0105). Polyclonal anti-phospho-SPT5 antibodies (pThr775) were raised by immunizing rabbits with a phosphorylated peptide (CGSQT(p)PMYGSGSRT(p)PMY) followed by negative affinity purification against the non-phosphorylated peptide described above, and subsequently by positive purification against the phosphorylated peptide.

For ChIP experiments, antibodies against pSer2 Pol II (Abcam, Cat. #ab5095), total Pol II (Santa Cruz, Cat. #sc-899 X), acetyl-histone H3 (Millipore, Cat. #06-599), acetyl-histone H4 (Millipore, Cat. #06-598) were purchased commercially. Antibodies against CDK9 and BRD4 were generated in our own laboratory and have been described previously (*Yang et al., 2001*).

All cell lines were obtained from American Type Culture Collection (ATCC, Manassas, VA) and grown in the media recommended by ATCC.

#### Measurement of selectivity profiles of i-CDK9

The inhibitory activity of i-CDK9 for CDK1-CycB (Upstate/Millipore), CDK2-CycA (Invitrogen), CDK4-CycD1 (Novartis), CDK7-CycH-MAT1 (Invitrogen), and CDK9-CycT1 (Invitrogen) was assessed in the AlphaScreen (PerkinElmer, Inc)-based kinase assay developed at Novartis. Briefly, each CDK–cyclin pair was pre-incubated with i-CDK9 for 15 min, followed by addition of kinase substrates (ELK for CDK1 and CDK2, RB for CDK4, and CDK7tide for CDK7 and CDK9). The mixture was incubated at room temperature for additional 4–5 hr. The reaction was stopped and read on Envision (PerkinElmer, Inc). Selectivity against a large panel of over 400 kinases was evaluated using the KINOMEscan platform (DiscoveRx, CA) employing a proprietary active site-directed competition binding assay (*Fabian et al., 2005*). The assay was conducted at two different concentrations of i-CDK9 (1 and 10 μM) by DiscoveRx.

#### ChIP-qPCR

The assay was performed as described (*Li et al., 2013*) with some modifications. Briefly, HeLa cells treated with DMSO or 0.3 μM i-CDK9 for 8 hr were cross-linked with 1% formaldehyde for 10 min. Cross-linking was quenched by the addition of glycine (0.125 M) for 5 min. Fixed cells were collected and resuspended in SDS lysis buffer (1% SDS, 10 mM EDTA, 50 mM Tris, pH 8.1). Chromatin DNA was fragmented with the Covaris-S2 sonicator (Covaris, Inc.) in 30 s on/30 s off cycles for a total processing time of 30 min. Sonicated lysates derived from $2 \times 10^6$ cells were incubated overnight with 3 μg the indicated antibodies per reaction and followed by an 1-hr incubation with Dynabeads Protein A (Life Technologies). Immunoprecipitated DNA was purified by PCR Purification Kit (QIAGEN) and analyzed by qPCR. The sequences of the PCR primers used for amplification of the various MYC regions are listed below:

A-Forward: AAGGCCTGGAGGCAGGAGTAATTT
A-Reverse: AGTTTGCAGCTCAGCGTTCAAGTG
B-Forward: ACGTTTGCGGGTTACATACAGTGC
B-Reverse: GAGAGGAGTATTACTTCCGTGCCT
C-Forward: GCGCGCCCATTAATACCCTTCTTT
C-Reverse: ATAAATCATCGCAGGCGGAACAGC
D-Forward: TACTCACAGGACAAGGATGCGGTT
D-Reverse: TGAATTAACTACGCGCGCCTACCA
E-Forward: TAGTAATTCCAGCGAGAGGCAGAG
E-Reverse: TATGGGCAAAGTTTCGTGGATGCG
F-Forward: ACTGGAACTTACAACACCCGAGCA
F-Reverse: TGGACTTCGGTGCTTACCTGGTTT
G-Forward: GGTTCACTAAGTGCGTCTCCGAGATA
G-Reverse: AGCGGGAGGCAGTCTTGAGTTAAA
H-Forward: TAAGGGTGGCTGGCTAGATTGGTT
H-Reverse: AAATTAGCCTGGCATGGTGGTGTG
I-Forward: ACAGAAATGTCCTGAGCAATCACC
I-Reverse: GCCCAAAGTCCAATTTGAGGCAGT
J-Forward: TTCCTCTGTTGAAATGGGTCTGGG
J-Reverse: ACCTGCCTTCTGCCATTCCTTCTA
K-Forward: TCCTGTCCATGGGTTATCTCGCAA
K-Reverse: AACAGAATGGGTCCAGATTGCTGC

All qPCR signals were normalized to the input DNA, and signals generated by nonspecific IgG in control immunoprecipitations were subtracted from the signals obtained with the specific antibodies.

## Co-immunoprecipitation

All Co-immunoprecipitation experiments were performed in NEs prepared from F1C2 cells that stably express CDK9-F or the parental HeLa cells transfected with the indicated cDNAs. The procedures have been described previously (*Lu et al., 2014*).

## Flow cytometry analysis

After incubation with the indicated compound(s), HeLa cells ($1 \times 10^6$) were washed twice with 1× phosphate buffered saline (PBS) and fixed in cold (−20°C) 70% ethanol overnight. To measure the sub-G1 population, cells were then stained with propidium iodide (50 µg/ml) for 1 hr at 37°C and analyzed by a Beckman Coulter Epics XL flow cytometer.

## In vitro kinase assay to measure the effects of BRD4 on CDK9 catalytic activity and response to i-CDK9

CDK9/CycT1 (Invitrogen) and purified, recombinant full-length or truncated BRD4 were pre-incubated for 45 min in a buffer containing 20 mM HEPES, pH 7.3, 10 mM $MgCl_2$, 0.01% Tween 20, 0.01% BSA and 1 mM DTT. i-CDK9 was either added or not added and the kinase reactions were immediately started by adding ATP and the CDK7tide peptide (NYSPTSPSYSPTSPSYSPTSPS; Bio-Synthesis, Inc.) substrate. The final reaction mixture contained 2 nM CDK9/cycT1, 50 nM CDK7tide, 1 mM ATP, 2.5% DMSO, and various concentrations of i-CDK9 plus or minus BRD4 as indicated in the figures. The reactions were stopped by adding 60 mM EDTA plus 0.01% Tween 20. After the addition of AlphaScreen beads (PerkinElmer) and detection antibody (Cell Signaling Technology, #4735) followed by overnight incubation at room temperature, the plates were read on an Envision 2101 (PerkinElmer).

## Isobologram to measure combined effects of i-CDK9 and JQ1 on cell growth and apoptosis

HeLa or H1792 cells were plated into 96-well microplates. i-CDK9 was serially diluted in the presence of seven different concentrations of JQ1. i-CDK9 or JQ1 was also serially diluted alone. The compounds were added to cells in 96-well dishes and incubated at 37°C for 24 hr for apoptosis assays or 72 hr for cell proliferation assays. Each treatment condition was done in triplicate. Apoptosis was detected by measuring Caspase 3/7 activation using Promega's Caspase-Glo 3/7 assay. Cell proliferation was measured using Promega's Celltiter-Glo assay. The concentrations of each drug in the combinations required to produce 50% inhibition were plotted to generate the isoboles.

Isobologram analysis of drug interaction was performed essentially as described (*Tallarida, 2001*; *Tse et al., 2007*).

## DNA microarray analysis

Cells were treated with DMSO or 0.3 µM i-CDK9 for 2, 8, and 16 hr, washed once with 1× PBS, and then lysed in 600 µl of RLT buffer that is part of the Qiagen RNeasy Mini Kit (QIAGEN), which was used to extract total RNA. RNA concentrations were measured on NanoDrop 2000 Spectrophotometer (Thermo Scientific) and RNA integrity determined by the Agilent RNA 6000 Nano Kit on Bioanalyzer (Agilent Technologies). 400 ng of total RNA from each sample that displayed good quality (RIN or RNA Integrity Number > 7) was submitted to the Novartis microarray core facility and analyzed on Affymetrix HGU133Plus2 microarray platform according to the manufacturer's instructions. Normalization of the raw signals was done using bioconductor affy package (http://bioconductor.org) with RMA method and HGU133Plus2_Hs_ENTREZG custom CDF for target gene definition (*Dai et al., 2005*). The metadata for the expression profiling of HeLa cells treated with i-CDK9 has been submitted to the NCBI GEO database and the accession number is GSE60952.

## ChIP-seq analysis

Upon treatment with i-CDK9, HeLa cells were fixed with 1% formaldehyde for 10 min, followed by the addition of 0.125 M glycine for 5 min to quench the reaction. The fixed cells were collected, re-suspended in SDS lysis buffer (1% SDS, 10 mM EDTA, 50 mM Tris, pH 8.1), and fragmented using Covaris-S2 sonicator (Covaris Inc.). Sonicated lysates were incubated with 5 µg Pol II antibody (sc-899x, Santa Cruz Biotechnology) overnight. After extensive washes, the isolated chromatin fraction was processed for sequencing library preparation according to the manufacture's instructions (Illumina Inc.). Sequencing was conducted by the Novartis sequencing core facility.

Raw reads were aligned using Bowtie v2.1.0 (*Langmead and Salzberg, 2012*) and ChIP-seq peaks were identified by running MACS v1.4.2 (*Zhang et al., 2008*) in treated vs control samples (non-default options: -B–nomodel–shiftsize 125). MACS also generated read pileups, which were used to compute the traveling ratio (*Rahl et al., 2010*). Specifically, Ref-Seq isoforms were used to compute the average read counts in a −30 bp to +300 bp window relative to the transcript start site (TSS). These were divided by the average read counts (TU) in the remaining gene body (+300 bp to end, including introns). To simplify analysis for regions with very low or zero read coverage, a small constant (BC; calculated as [total read counts × read lengths]/genome size) that represents the average coverage genome-wide was also added in the calculation. TR was thus computed as TR = (TSS + BC)/(TU + BC). Isoforms were excluded from this analysis if they were shorter than 650 bp or if the interval of 30 bp upstream to end of gene didn't overlap with any of the MACS peaks called for that sample. Gene-level TR was computed by averaging the TRs of all isoforms of that gene. Pol II coverage of selected genes was visualized using custom python scripts (*Figure 1—source data 1*) to plot normalized coverage over entire gene length plus a margin that equal to 7% of the gene length on either end. The metadata for the Pol II occupancy profiling by ChIP-seq in HeLa cells in the presence and absence of i-CDK9 has been submitted to the NCBI GEO database and the accession number is GSE60953.

## Gene set and GO enrichment analysis

GSEA analysis was carried out as specified by Broad GSEA package using default parameters (*Subramanian et al., 2005*). We used DAVID bioinformatics resources (*Huang da et al., 2009*) to evaluate enrichment of GO biological processes (GOTERM_BP_FAT) impacted by CDK9 inhibition, using top 500 genes that were down-regulated by CDK9 inhibitor treatment, or top 500 genes that had increased TR upon CDK9 inhibition. FDR was used to rank the GO terms following gene list search.

## Acknowledgements

We thank Bob Warne for technical support in performing the biochemical kinase assay, Lorraine Kent for support in the cell-based assay, Dr Matija Peterlin for CDK12 expression vector, and Caleb Chan for support in the ChIP assay. This work is supported by grants (R01AI41757 and R01AI095057) from the National Institutes of Health and a grant (W81XWH-15-1-0067) from the Department of Defense to QZ. It is also supported by a grant (W81XWH-15-1-0068) from the Department of Defense to KL.

## Additional information

### Funding

| Funder | Grant reference | Author |
|---|---|---|
| National Institutes of Health (NIH) | R01AI41757 | Qiang Zhou |
| National Institutes of Health (NIH) | R01AI095057 | Qiang Zhou |
| U.S. Department of Defense | W81XWH-15-1-0067 | Qiang Zhou |
| U.S. Department of Defense | W81XWH-15-1-0068 | Kunxin Luo |
| National Natural Science Foundation of China | 81201276 | Yuhua Xue |

The funders had no role in study design, data collection and interpretation, or the decision to submit the work for publication.

### Author contributions

HL, Conception and design, Acquisition of data, Analysis and interpretation of data, Drafting or revising the article; YX, Conception and design, Acquisition of data; GKY, BW, Analysis and interpretation of data, Drafting or revising the article; CA, Acquisition of data, Analysis and interpretation of data; JL, SF, MF, AM, JS, Analysis and interpretation of data, Contributed unpublished essential data or reagents; XJ, Acquisition of data, Analysis and interpretation of data, Drafting or revising the article; KL, Conception and design, Analysis and interpretation of data; ZG, QZ, Conception and design, Analysis and interpretation of data, Drafting or revising the article

## Additional files

### Major datasets

The following datasets were generated:

| Author(s) | Year | Dataset title | Dataset ID and/or URL | Database, license, and accessibility information |
|---|---|---|---|---|
| Yu GK, Lu H, Gao Z, Zhou Q | 2015 | Expression profiling of HeLa cell line treated with CDK9 inhibitor | www.ncbi.nlm.nih.gov/geo/query/acc.cgi?GSE60952 | Publicly available at the NCBI Gene Expression Omnibus (Accession no: GSE60952). |
| Yu GK, Arias C, Weisburd B, Mercier A, Lu H, Zhou Q | 2015 | Compensatory induction of MYC expression by sustained CDK9 inhibition via a BRD4-dependent mechanism: Pol II occupancy profiling by ChIP-Seq in HeLa cell line in the presence or absence of CDK9 inhibition | www.ncbi.nlm.nih.gov/geo/query/acc.cgi?GSE60953 | Publicly available at the NCBI Gene Expression Omnibus (Accession no: GSE60953). |

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
