## [Decision Letter]

Thank you for sending your work entitled “Compensatory induction of MYC expression by sustained CDK9 inhibition via a BRD4-dependent mechanism” for consideration at *eLife*. Your article has been evaluated by James Manley (Senior editor) and three reviewers, one of whom is a member of our Board of Reviewing Editors.

The Reviewing editor and the other reviewers discussed their comments before we reached this decision, and the Reviewing editor has assembled the following comments to help you prepare a revised submission.

After extensive consultation, the reviewers recommend that this paper be revised with new experiments, and the following is a summary prepared by the Reviewing Editor. It should be noted that the very positive Reviewer 1 was least knowledgeable about the literature and relevant issues and was convinced by the other reviewers of the weaknesses of the paper. These weaknesses include both experimental issues and also the relative lack of mechanism (which is why the main phenomena have to be addressed experimentally). As such, the reviewers are all in accord with this decision.

The authors should address all the comments. In particular, experiments that should be done include:

1) Specificity of the inhibitor against CDK12.

2) A key point to address is a direct test of whether transcription induced by i-CDK9 is dependent on the catalytic activity of CDK9. This would require looking at higher concentrations of the inhibitor and to achieve induction first and then try to shut off with the higher dose.

3) Analysis at earlier time-points. The question is whether there is anything different in the chromatin of a transiently repressed gene (MYC) versus a “permanently” repressed one. ChIPs of histone acetylation and BRD4 at 2 hr at minimum (although patterns of histone methylation might also be informative).

4) The Keskin paper and the paper about DYRK1a mentioned in review (Di Vona et al., Mol. Cell 57:506-20, 2015) need to be cited. There is some literature on related effects of some genes being activated upon inhibition of CDK9

Reviewer #1:

This is a superb paper that should be published in *eLife*. The authors describe a selective inhibitor of CDK9, and the evidence for selectivity is excellent (of course, not perfect). While, as expected, CDK9 inhibition generally inhibits Pol II release from the pause, they show that MYC and some other genes behave in the opposite fashion. They convincingly show that CDK9 inhibition leads to release of CDK9 from the 7SK snRNA, increased association with BRD4 leading to enhanced CDK9 kinase activity and resistance to inhibition. In addition, they show that increased MYC expression compensates for reduced CDK9 function and that simultaneous inactivation of MYC and CDK9 synergistically induces growth arrest. Taken together, this paper describes a new and potentially therapeutically useful inhibitor of CDK9, as well as a novel mechanistic and biological connection between MYC and CDK9. The paper is very well written.

Reviewer #2:

The authors describe a new inhibitor of cyclin-dependent kinase 9 (CDK9), with improved potency and selectivity compared to drugs previously available. At the majority of genes, treatment of cells with this compound, i-CDK9, has the predicted effects on Pol II distribution-increased promoter-proximal pausing and decreased occupancy over gene bodies-and on steady-state transcript levels, which are decreased. At a subset of genes, however, CDK9 inhibition had little effect on pausing (measured as the “traveling ratio” or TR), and caused expression levels to increase. Among the genes that responded paradoxically was MYC, a key growth-promoting transcription factor. After transient repression by i-CDK9, levels of MYC transcription recover and then surpass levels in DMSO-treated cells. The authors show that i-CDK9 treatment causes 1) release of CDK9 from inhibitory 7SK RNA/HEXIM1 complexes; 2) increased association with the stimulatory/targeting partner BRD4; 3) enhanced occupancy of Pol II, BRD4 and CDK9 at the MYC gene; and 4) increased phosphorylation at Ser2 of the Pol II CTD on MYC. Paradoxically, increased expression of MYC induced by i-CDK9 appears to depend on CDK9 activity.

The experiments are mostly well executed and interpreted. CDK9/cyclin T1, a.k.a. P-TEFb, is a central regulator of gene expression and, possibly, a cancer drug target. A truly selective inhibitor would be of great interest to investigators in the field. The demonstration that MYC can evade repression by i-CDK9 is interesting, but not entirely novel; differential sensitivity of specific genes to inhibitors of “general” transcriptional functions has many precedents. This is MYC, so the mechanism of this evasion will be of significant interest to a wide audience, but the results presented here do not provide that mechanistic understanding. Why certain subpopulations of the released CDK9 would be resistant to such a potent inhibitor, or why certain loci might have an advantage in recruiting transcription machinery, including CDK9, under those conditions, is left unexplained. Extensive additional experiments would be needed to raise the work to the level at which I could recommend it for publication in *eLife*. My specific concerns are listed below:

1) As stated above, no mechanism is provided to explain the most interesting effects of i-CDK9 on gene expression. I think the authors make a tactical error by choosing to focus nearly exclusively on the late (8-hr) time point when MYC transcription is maximally induced. The real clues to how MYC is being “re-activated” might be found by asking why the initial repression seen at 2 hr does not persist (in comparison with genes that are durably repressed by i-CDK9). Are there differences in the occupancy of transcription machinery at those genes? In chromatin modification patterns? For example, increased histone acetylation on MYC at later time points is not surprising, because transcription is elevated (Figure 5), but analyzing such marks at 2 hr might be more informative as to the mechanism of activation.

2) The ability of “transcriptional stress” (including treatments with drugs that target CDK9) to promote the release of CDK9 from 7SK/HEXIM1 is well-known. The authors' analysis of this phenomenon is more detailed than most, but does not illuminate the mechanism, i.e., how lowered CDK9 activity feeds back on CDK9-regulatory pathways. It also raises another question: for all the apparent superiority of i-CDK9 to older drugs (lower IC50, fewer and less sensitive secondary targets), most of the results presented here could have (or have) been obtained with flavopiridol at similar doses. This extends to effects seen at cellular (PARP cleavage), genomic (increased TR) and biochemical levels (decreased Ser2 and Spt5 phosphorylation).

3) A paper just published in Molecular Cell (February 5, so probably after this manuscript was submitted), proposes a novel function for DYRK1A-a secondary target of i-CDK9 with an IC50 ∼50 nM-as a Ser2- and Ser5-specific CTD kinase. I do not expect the authors to respond to this paper, but they should acknowledge that it exists and could explain some of the effects of i-CDK9 on Ser2 phosphorylation. It is quite likely, in my opinion, that no single kinase is responsible for any specific phosphorylation on Pol II in human cells, so whenever one sees disappearance of pSer2 (e.g. at i-CDK9 doses > 300 nM in Figure 1), it is likely to signify inhibition of multiple kinases. The flip side is that effects seen at lower doses are potentially due to incomplete inhibition of the intended target. This rather than the modest effects of BRD4-binding on CDK9 activity-the ∼2-fold stimulation of kinase (Figure 4) and ∼3-fold increase in IC50 (Figure 4)-is probably why MYC expression can be induced by i-CDK9 even though it depends on CDK9 activity. I also disagree with the assertion (in the first paragraph of the subsection headed “Binding of BRD4 to P-TEFb increases CDK9’s catalytic activity and resistance to inhibition”) that MYC induction is happening “before” CDK9 inactivation, because it rests on a false assumption: that decay of phospho-Ser2 and -Spt5 signals provides an instantaneous measurement of kinase activity in vivo. That is certainly not the case; even if no other kinase can phosphorylate these sites, the loss of phosphates will depend on activity of the phosphatases that remove them. To clarify the situation, I would ask a simple question: can MYC expression induced by 100-500 nM i-CDK9 be shut off by subsequent addition of the same drug at a higher dose?

4) The ChIP data in Figure 5 are consistent with the elevated MYC expression at the 8-hr time point. BRD4, CDK9 and Pol II are all increased, as expected. Why does the increase in CDK9 appear proportionately greater than that of BRD4, however, if the latter is needed to recruit the former? The authors also make the point that the pSer2 distribution is shifted towards the 3' end relative to that of CDK9, but this is consistent with many published results. What looks more noteworthy is the apparent shift of pSer2 towards the 5' end in the i-CDK9-treated cells relative to the DMSO-treated ones. To be sure that this is real, they would need to plot the pSer2:total Pol II ratio (which they should do in any case).

Reviewer #3:

The authors report a new chemical inhibitor of Cdk9, which is more potent and selective than flavopiridol, but has similar effects on the cell. They show that these compounds have effects on cells that are not seen upon knockdown of Cdk9, namely increased expression of MYC, FOS, JUN and other Brd4-sensitive genes. The findings suggest a positive feedforward loop in which cells respond to low levels of Cdk9 by increasing Brd4-binding and activation of P-TEFb.

1) Throughout the paper, global RNA pol II Ser2 phosphorylation is used as a marker for Cdk9 kinase activity. However, it has been reported that Cdk9 knockdown does not affect the global level of Ser2 phosphorylation, and instead that Cdk12 (yeast Ctk1) is the CTD Ser2-kinase ([4], Genes Dev 15, 2303-16). So the authors should show that the knockdown of Cdk9 causes a loss of Ser2 phosphorylation in their hands. Otherwise, the effect of i-Cdk9 on Ser2 phosphorylation could be due to inhibiting a different kinase.

2) The very related question is whether i-Cdk9 inhibits Cdk12? The authors show that i-Cdk9 has fewer off-target effects than flavopiridol, but this paper ignores whether or not i-Cdk9 affects Cdk12 kinase activity. Flavopiridol is a significant inhibitor of Cdk12, and if i-Cdk9 also blocks Cdk12, it would change the conclusions of the paper, and certainly invalidate using Ser2 phosphorylation as a marker for Cdk9 because Cdk12 is critical for cancer cell survival, the 'anti-cancer' effects of flavopiridol, and possibly also i-Cdk9, could have more to do with inhibition of Cdk12 than Cdk9.

3) The CTD Ser2 phosphorylation increases in the body of the MYC gene upon treatment with i-Cdk9, consistent with increased transcription of the gene. However, eventually the MYC transcription and CTD Ser2-P should be inhibited by i-Cdk9, at extended times or with higher levels of the drug? To establish the correlation with Brd4, it needs to be shown that CTD Ser2 levels go down under the same conditions at non-Brd-dependent genes.

[Editors' note: further revisions were requested prior to acceptance, as described below.]

Thank you for resubmitting your work entitled “Compensatory induction of MYC expression by sustained CDK9 inhibition via a BRD4-dependent mechanism” for further consideration at *eLife*. Your revised article has been evaluated by James Manley (Senior editor), a Reviewing editor, and two reviewers. The manuscript has been improved but there are some remaining issues that need to be addressed before acceptance, as outlined below:

Reviewer 1 strongly suggested a low dose-high dose experiment, but is willing to have the paper published provided if it is acknowledged in the text the drawbacks/pitfalls of both the knockdown and dominant-negative approaches, leaving open the possibility of other interpretations. If this change in the text is done appropriately, the paper will be accepted.

Reviewer #1:

This is a revised version of a manuscript I previously reviewed. It is improved, most significantly by the demonstration that changes in chromatin modifications and BRD4 recruitment/ retention precede transcriptional induction of MYC and other genes that respond paradoxically to the CDK9 inhibitor (Figure 5). The authors also demonstrate that CDK12 is inhibited only very weakly by i-CDK9. The remaining major concern is the decision not to do the experiment I suggested: first to induce MYC expression by low-dose (e.g. 300 nM) i-CDK9 treatment for 8 hr, and then try to shut it off with a higher dose (2 µM). The response to this suggestion, both in the revised manuscript (in the last paragraph of the subsection headed “The kinase-active P-TEFb and its interaction with BRD4 are required for i-CDK9 to induce MYC expression”) and in the rebuttal letter, is to restate results that were in the original version, which I understood but found unsatisfactory. Here's why:

1) I know that the D167N mutant acts dominant-negatively, but that is the problem. Its ability to bind CDK9 partners, ATP and i-CDK9 means that overexpressing it is likely to change the response to the drug, without necessarily showing that CDK9 activity is required for MYC transcriptional induction (i.e., it could just be “soaking up” drug).

2) The same problem applies in reverse to the RNAi experiment: silencing CDK9 expression depletes the primary target of i-CDK9, before the drug is added. If induction of MYC expression by i-CDK9 is through binding to CDK9, it is hard to see how losing this effect when CDK9 is depleted says anything about a catalytic requirement for CDK9 in induced MYC transcription.

3) Finally, the fact that high-dose i-CDK9 represses MYC when added at time 0 does not say that the same treatment would shut off transcription induced by the lower dose. The confusion is evident in the sentence “The dependence on catalytically active P-TEFb […] higher i-CDK9 concentration”, which uses the language of a time course (“initially induced,” “ultimately suppressed”) to describe a simple dose response (subsection “The kinase-active P-TEFb and its interaction with BRD4 are required for i-CDK9 to induce MYC expression”). I want to see the actual time course, in cells that are first treated with the lower dose to induce MYC expression and then raised to the higher dose to try to shut it off. From that result I would conclude that MYC transcription induced by i-CDK9 still depended on CDK9 activity.

---

## [Author Response]

*1) Specificity of the inhibitor against CDK12*.

To determine how i-CDK9 may affect CDK12 kinase activity, we compared the abilities of CDK12 and CDK9 to phosphorylate GST-CTD in the presence of increasing amounts of i-CDK9. Phosphorylation of Pol II CTD on Ser2 was detected by Western blotting with the pSer2-specific antibody. Two different sources of materials were tested in separate experiments in order to ensure the consistency of the observations. First, Flag-tagged CDK12 (CDK12-F) and CDK9 (CDK9-F) affinity-purified from transfected HeLa cells were tested in kinase reactions shown in new Figure 1. While CTD phosphorylation by CDK9-F was efficiently inhibited by i-CDK9 with an estimated IC50 of 2 nM under the current experimental conditions, no obvious inhibition of CDK12-F was detected even at 80 nM of the drug. Next, the sensitivity of the two kinases to i-CDK9 was also compared in kinase reactions containing the baculovirus-produced recombinant CDK9-CycT1 and CDK12-CycK (SignalChem). As shown in new Figure 1—figure supplement 1, while Ser2 phosphorylation by CDK9-CycT1 was mostly inhibited by 80 nM i-CDK9, CDK12-CycK was not significantly inhibited until 640-1,200 nM i-CDK9 was added into the reactions (Figure 1—figure supplement 1). Thus, between CDK9 and CDK12, i-CDK9 displays high selectivity against the former.

It’s interesting to note that a similar finding was recently made by Dr. Geyer and colleagues, who showed that compared with CDK9, CDK12 was about one order of magnitude less sensitive to inhibition by the pan-CDK inhibitor flavopiridol (8).

*2) A key point to address is a direct test of whether transcription induced by i-CDK9 is dependent on the catalytic activity of CDK9. This would require looking at higher concentrations of the inhibitor and to achieve induction first and then try to shut off with the higher dose*.

The i-CDK9-induced MYC expression is indeed dependent on the catalytic activity of CDK9. This point has already been clearly demonstrated in Figure 4, which shows that the overexpression of a dominant-negative, kinase-dead CDK9 mutant (D167N) effectively blocked the induction of MYC expression by i-CDK9. As for the suggested experiment to look at MYC expression in response to higher concentrations of i-CDK9, it has also been done in Figure 7, which shows that MYC expression was initially induced by 0.3-0.5 mM i-CDK9 but significantly suppressed to below the basal level by 2 mM i-CDK9 in control cells expressing shGFP. These results and their implication of the dependence on CDK9’s catalytic activity for MYC induction are now strongly emphasized in the last paragraph of the subsection headed “The kinase-active P-TEFb and its interaction with BRD4 are required for i-CDK9 to induce 346 MYC expression”.

*3) Analysis at earlier time-points. The question is whether there is anything different in the chromatin of a transiently repressed gene (MYC) versus a* “*permanently*” *repressed one. ChIPs of histone acetylation and BRD4 at 2 hr at minimum (although patterns of histone methylation might also be informative)*.

Following the reviewers’ suggestion, we performed ChIP analyses of H3 and H4 acetylation and BRD4 binding at the promoters of MYC, a transiently repressed gene, and HEXIM1, an example of a “permanently” repressed gene (see Figure 3—figure supplement 2), at early time points (0, 1 and 2 hr) of the i-CDK9 treatment. As shown in new Figure 5, upon exposure to i-CDK9, the levels of both Ac-H3 and Ac-H4 at the MYC promoter region began to increase, with the more robust increase observed for Ac-H4. In contrast, at the HEXIM1 gene promoter, i-CDK9 caused a drastic decrease in the Ac-H3 level but a small increase of the Ac-H4 level. As for the BRD4 level, it displayed a marked decrease at the MYC promoter at 1 hr post i-CDK9 treatment, but began to rebound by 2 hr. At the HEXIM1 promoter, however, it showed a sustained reduction throughout the entire period.

Thus, at early time-points of i-CDK9 treatment before MYC transcription is induced, the acetylation state of the MYC promoter is already different from that of the HEXIM1 promoter. It is possible that the simultaneous increase in both Ac-H3 and Ac-H4 levels detected at the MYC promoter is a good prediction of eventual “re-activation” of the gene. In contrast, a strong reduction in Ac-H3 level that cannot be offset by a weak Ac-H4 increase as observed at the HEXIM1 promoter could be an indicator of a permanently repressed state caused by i-CDK9.

4) The Keskin paper and the paper about DYRK1a mentioned in review (Di Vona et al., Mol. Cell 57:506-20, 2015) need to be cited. There is some literature on related effects of some genes being activated upon inhibition of CDK9

We thank the reviewers for pointing out the two papers, which have now been discussed in the revised manuscript. For the Keskin et al. paper, we discussed (in the second paragraph of the Discussion section) that just like what we are showing with i-CDK9 in the present study, a subset of the same primary response genes that were examined in our Figure 6 have also been shown by Keskin et al. to display a very similar biphasic response (initial down-regulation followed by up-regulation) following the treatment with the pan-CDK inhibitor flavopiridol.

For the DYRK1a paper by Di Vona et al., we discussed (in the last paragraph of the subsection headed “i-CDK9 is a potent and selective CDK9 inhibitor”) that despite a more than 100-fold greater selectivity of i-CDK9 for CDK9 than for DYRK1A ([Supplementary-material SD1-data]), the fact that the latter is one of the best targets of i-CDK9 after CDK9 in our inhibition assay suggests that the structure and/or function of DYRK1A may resemble that of CDK9 to a certain degree. In light of this similarity, it is not surprising to see that DYRK1A has recently been found to act as a Pol II CTD Kinase at its target gene promoters (15).

Reviewer #2:

*1) As stated above, no mechanism is provided to explain the most interesting effects of i-CDK9 on gene expression. I think the authors make a tactical error by choosing to focus nearly exclusively on the late (8-hr) time point when MYC transcription is maximally induced. The real clues to how MYC is being* “*re-activated*” *might be found by asking why the initial repression seen at 2 hr does not persist (in comparison with genes that are durably repressed by i-CDK9). Are there differences in the occupancy of transcription machinery at those genes? In chromatin modification patterns? For example, increased histone acetylation on MYC at later time points is not surprising, because transcription is elevated (*Figure 5*), but analyzing such marks at 2 hr might be more informative as to the mechanism of activation.*

Please see our response above to common concern #3 shared by all reviewers.

*2) The ability of* “*transcriptional stress*” *(including treatments with drugs that target CDK9) to promote the release of CDK9 from 7SK/HEXIM1 is well-known. The authors' analysis of this phenomenon is more detailed than most, but does not illuminate the mechanism, i.e., how lowered CDK9 activity feeds back on CDK9-regulatory pathways. It also raises another question: for all the apparent superiority of i-CDK9 to older drugs (lower IC50, fewer and less sensitive secondary targets), most of the results presented here could have (or have) been obtained with flavopiridol at similar doses. This extends to effects seen at cellular (PARP cleavage), genomic (increased TR) and biochemical levels (decreased Ser2 and Spt5 phosphorylation).*

We are not quite sure about what the reviewer wants us to improve here. With regard to the mechanism and signaling pathway that regulate the release of P-TEFb from 7SK snRNP in response to “transcriptional stress”, they indeed remain mostly unknown at this point, but are not the focus of the current study. There have been several reports on how other types of stress signals (e.g. UV and HMBA) cause the release of P-TEFb, and it is possible that similar mechanisms and signaling molecules may play a role in allowing the “transcriptional stress” to do the same thing.

*3) A paper just published in Molecular Cell (February 5, so probably after this manuscript was submitted), proposes a novel function for DYRK1A-a secondary target of i-CDK9 with an IC50 ∼50 nM-as a Ser2- and Ser5-specific CTD kinase. I do not expect the authors to respond to this paper, but they should acknowledge that it exists and could explain some of the effects of i-CDK9 on Ser2 phosphorylation. It is quite likely, in my opinion, that no single kinase is responsible for any specific phosphorylation on Pol II in human cells, so whenever one sees disappearance of pSer2 (e.g. at i-CDK9 doses > 300 nM in*
Figure 1*), it is likely to signify inhibition of multiple kinases. The flip side is that effects seen at lower doses are potentially due to incomplete inhibition of the intended target. This rather than the modest effects of BRD4-binding on CDK9 activity-the ∼2-fold stimulation of kinase (*Figure 4*) and ∼3-fold increase in IC50 (*Figure 4*)-is probably why MYC expression can be induced by i-CDK9 even though it depends on CDK9 activity. I also disagree with the assertion (in the first paragraph of the subsection headed “Binding of BRD4 to P-TEFb increases CDK9’s catalytic activity and resistance to inhibition”) that MYC induction is happening* “*before*” *CDK9 inactivation, because it rests on a false assumption: that decay of phospho-Ser2 and -Spt5 signals provides an instantaneous measurement of kinase activity in vivo. That is certainly not the case; even if no other kinase can phosphorylate these sites, the loss of phosphates will depend on activity of the phosphatases that remove them. To clarify the situation, I would ask a simple question: can MYC expression induced by 100-500 nM i-CDK9 be shut off by subsequent addition of the same drug at a higher dose?*

The recent MC paper by Di Vona et al. has been discussed in the last paragraph of the subsection headed “i-CDK9 is a potent and selective CDK9 inhibitor” of the revised manuscript (please also see our response above to common concern #4 shared by all three reviewers). Regarding the suggested experiment to examine MYC expression in the presence of high concentrations of i-CDK9, it has already been done in Figure 7, which shows that MYC expression was initially induced by 0.3-0.5 mM of i-CDK9 but significantly down-regulated to even below the basal level by 2 mM of i-CDK9 in control HeLa cells expressing shGFP. This result, together with the data employing the dominant-negative CDK9 mutant D167N, confirms that the i-CDK9-induced MYC expression is indeed dependent on the catalytic activity of CDK9.

*4) The ChIP data in*
Figure 5
*are consistent with the elevated MYC expression at the 8-hr time point. BRD4, CDK9 and Pol II are all increased, as expected. Why does the increase in CDK9 appear proportionately greater than that of BRD4, however, if the latter is needed to recruit the former? The authors also make the point that the pSer2 distribution is shifted towards the 3' end relative to that of CDK9, but this is consistent with many published results. What looks more noteworthy is the apparent shift of pSer2 towards the 5' end in the i-CDK9-treated cells relative to the DMSO-treated ones. To be sure that this is real, they would need to plot the pSer2:total Pol II ratio (which they should do in any case)*.

Because CDK9 and BRD4 were recognized by their own specific antibodies in the ChIP assay, the somewhat bigger increase of the CDK9 signal compared to that of BRD4 may simply reflect the different dynamic ranges of recognition or sensitivities of the two antibodies. Based on the reviewer’s suggestion, we have plotted the pSer2/total Pol II ratios across the MYC locus under the DMSO and i-CDK9 conditions and the result is shown in Figure 8. The ratio is uniformly higher across the entire MYC locus except for the extreme 3’ position (K) following the treatment with 0.3 mM i-CDK9 for 8 hr. This is consistent with the enhanced occupancy of the BRD4-P-TEFb complex at the locus and elevated transcriptional elongation under these conditions. However, no obvious shift of pSer2 toward the 5’ end was detected in the plot.

Author response image 1.**DOI:**
http://dx.doi.org/10.7554/eLife.06535.024

Reviewer #3:

*1) Throughout the paper, global RNA pol II Ser2 phosphorylation is used as a marker for Cdk9 kinase activity. However, it has been reported that Cdk9 knockdown does not affect the global level of Ser2 phosphorylation, and instead that Cdk12 (yeast Ctk1) is the CTD Ser2-kinase (*[4]*, Genes Dev 15, 2303-16). So the authors should show that the knockdown of Cdk9 causes a loss of Ser2 phosphorylation in their hands. Otherwise, the effect of i-Cdk9 on Ser2 phosphorylation could be due to inhibiting a different kinase*.

We thank the reviewer for giving us an opportunity to explain this perplexing controversy. Indeed, in our own hands, knockdown of both CDK9 and CycT1 (∼10% CDK9 and 50% CycT1 remained in KD cells) caused only a partial reduction of the global pSer2 level (Figure 9 Panel A; ∼30% remaining). However, when we looked at what was happening to the remaining CDK9 in the KD cells by measuring the signature 7SK snRNP component HEXIM1 and the signature Super Elongation Complex (SEC) components AFF4 and ELL2 that were associated with the immunoprecipitated CDK9 (Figure 9 Panel B), we think we know the reason behind the partial pSer2 reduction. When the CDK9 signals were adjusted to a similar level (the CDK9 level in the KD cells was very low; and we had to use many times more the KD cells than the control in order to get a comparable CDK9 signal), it is obvious from the co-IP/Western analysis (Panel B) that there was a dramatic shift of P-TEFb from the 7SK snRNP to the SEC under the KD conditions. Since the vast majority of P-TEFb (up to 90% by some estimations) is normally sequestered in the 7SK snRNP, this shift basically has converted all the remaining P-TEFb into the active SEC complex. Thus, although the KD has significantly decreased the overall P-TEFb level in vivo, it failed to completely eliminate the active form of P-TEFb due to the drastic P-TEFb mobilization that was happening inside the cell, which allowed the detection of still a decent level of global pSer2 under these conditions.

Author response image 2.**DOI:**
http://dx.doi.org/10.7554/eLife.06535.025

*2) The very related question is whether i-Cdk9 inhibits Cdk12? The authors show that i-Cdk9 has fewer off-target effects than flavopiridol, but this paper ignores whether or not i-Cdk9 affects Cdk12 kinase activity. Flavopiridol is a significant inhibitor of Cdk12, and if i-Cdk9 also blocks Cdk12, it would change the conclusions of the paper, and certainly invalidate using Ser2 phosphorylation as a marker for Cdk9 because Cdk12 is critical for cancer cell survival, the 'anti-cancer' effects of flavopiridol, and possibly also i-Cdk9, could have more to do with inhibition of Cdk12 than Cdk9*.

Please see our response above to common concern #1 shared by all reviewers.

*3) The CTD Ser2 phosphorylation increases in the body of the MYC gene upon treatment with i-Cdk9, consistent with increased transcription of the gene. However, eventually the MYC transcription and CTD Ser2-P should be inhibited by i-Cdk9, at extended times or with higher levels of the drug? To establish the correlation with Brd4, it needs to be shown that CTD Ser2 levels go down under the same conditions at non-Brd-dependent genes*.

Please see our response above to common concern #2 shared by all reviewers. Indeed, as shown in Figure 7, both the MYC expression and CTD pSer2 were eventually inhibited by a high concentration (2 mM) of i-CDK9. Consistent with the “permanent” inhibition of transcription of the HEXIM1 gene, treatment with 0.3 mM i-CDK9 for 8 hr, a condition that induced MYC transcription and increased the pSer2 level in the body of the MYC gene (Figure 5), caused a reduction of both total Pol II and the Ser2-phosphorylated Pol II across the entire HEXIM1 locus (new Figure 5—figure supplement 2).

[Editors' note: further revisions were requested prior to acceptance, as described below.]

*Reviewer 1 strongly suggested a low dose-high dose experiment, but is willing to have the paper published provided if it is acknowledged in the text the drawbacks/pitfalls of both the knockdown and dominant-negative approaches, leaving open the possibility of other interpretations. If this change in the text is done appropriately, the paper will be accepted*.

*Reviewer #1*:

*This is a revised version of a manuscript I previously reviewed. It is improved, most significantly by the demonstration that changes in chromatin modifications and BRD4 recruitment/ retention precede transcriptional induction of MYC and other genes that respond paradoxically to the CDK9 inhibitor (*Figure 5*). The authors also demonstrate that CDK12 is inhibited only very weakly by i-CDK9. The remaining major concern is the decision not to do the experiment I suggested: first to induce MYC expression by low-dose (e.g. 300 nM) i-CDK9 treatment for 8 hr, and then try to shut it off with a higher dose (2 µM). The response to this suggestion, both in the revised manuscript (in the last paragraph of the subsection headed “The kinase-active P-TEFb and its interaction with BRD4 are required for i-CDK9 to induce MYC expression”) and in the rebuttal letter, is to restate results that were in the original version, which I understood but found unsatisfactory. Here's why*:

*1) I know that the D167N mutant acts dominant-negatively, but that is the problem. Its ability to bind CDK9 partners, ATP and i-CDK9 means that overexpressing it is likely to change the response to the drug, without necessarily showing that CDK9 activity is required for MYC transcriptional induction (i.e., it could just be* “*soaking up*” *drug)*.

*2) The same problem applies in reverse to the RNAi experiment: silencing CDK9 expression depletes the primary target of i-CDK9, before the drug is added. If induction of MYC expression by i-CDK9 is through binding to CDK9, it is hard to see how losing this effect when CDK9 is depleted says anything about a catalytic requirement for CDK9 in induced MYC transcription*.

*3) Finally, the fact that high-dose i-CDK9 represses MYC when added at time 0 does not say that the same treatment would shut off transcription induced by the lower dose. The confusion is evident in the sentence* “*The dependence on catalytically active P-TEFb […] higher i-CDK9 concentration*”*, which uses the language of a time course (*“*initially induced,*” “*ultimately suppressed*”*) to describe a simple dose response (subsection “The kinase-active P-TEFb and its interaction with BRD4 are required for i-CDK9 to induce MYC expression”). I want to see the actual time course, in cells that are first treated with the lower dose to induce MYC expression and then raised to the higher dose to try to shut it off. From that result I would conclude that MYC transcription induced by i-CDK9 still depended on CDK9 activity*.

We have performed the so-called low dose (0.3 μM)-high dose (2 μM) experiment as suggested by Reviewer #1, and the result is the same as what was shown in Figure 7, in which case different amounts of i-CDK9 were added at time zero. The new result, which is now displayed in Figure 4—figure supplement 4 after the reviewer’s strong insistence, indeed shows that the induced MYC expression by 0.3 μM i-CDK9 can be subsequently shut off by 2 μM of the drug.